# PARALLELBENCH: UNDERSTANDING THE TRADE-OFFS OF PARALLEL DECODING IN DIFFUSION LLMS

**Wonjun Kang**[*1,5]    **Kevin Galim**[*1]    **Seunghyuk Oh**[*1]    **Minjae Lee**[1]
**Yuchen Zeng**[2,3]    **Shuibai Zhang**[2]    **Coleman Hooper**[4]    **Yuezhou Hu**[4]
**Hyung Il Koo**[1]    **Nam Ik Cho**[5]    **Kangwook Lee**[2,6,7]

[1] FuriosaAI    [2] UW-Madison    [3] Microsoft Research    [4] UC Berkeley
[5] Seoul National University    [6] KRAFTON    [7] Ludo Robotics

Project Page: https://parallelbench.github.io
Leaderboard: https://parallelbench.github.io/leaderboard/
GitHub: https://github.com/furiosa-ai/ParallelBench

## ABSTRACT

While most autoregressive LLMs are constrained to one-by-one decoding, diffusion LLMs (dLLMs) have attracted growing interest for their potential to dramatically accelerate inference through parallel decoding. Despite this promise, the conditional independence assumption in dLLMs causes parallel decoding to ignore token dependencies, inevitably degrading generation quality when these dependencies are strong. However, existing works largely overlook these inherent challenges, and evaluations on standard benchmarks (e.g., math and coding) are not sufficient to capture the quality degradation caused by parallel decoding. To address this gap, we first provide an information-theoretic analysis of parallel decoding. We then conduct case studies on analytically tractable synthetic list operations from both data distribution and decoding strategy perspectives, offering quantitative insights that highlight the fundamental limitations of parallel decoding. Building on these insights, we propose **PARALLELBENCH**, the first benchmark specifically designed for dLLMs, featuring realistic tasks that are trivial for humans and autoregressive LLMs yet exceptionally challenging for dLLMs under parallel decoding. Using PARALLELBENCH, we systematically analyze both dLLMs and autoregressive LLMs, revealing that: (i) dLLMs under parallel decoding can suffer dramatic quality degradation in real-world scenarios, and (ii) current parallel decoding strategies struggle to adapt their degree of parallelism based on task difficulty, thus failing to achieve meaningful speedup without compromising quality. Our findings underscore the pressing need for innovative decoding methods that can overcome the current speed-quality trade-off. We release our benchmark to help accelerate the development of truly efficient dLLMs.

## 1 INTRODUCTION

Large Language Models (LLMs) (Brown et al., 2020; Touvron et al., 2023) have achieved remarkable success in natural language processing, including complex reasoning (Jaech et al., 2024; Guo et al., 2025) and code generation (Jiang et al., 2024), which require extensive generation lengths. However, the autoregressive nature of most current LLMs fundamentally constrains inference speed, as they must generate tokens one-by-one. Unlike autoregressive LLMs, which follow (i) one-by-one decoding and (ii) left-to-right decoding, diffusion LLMs (dLLMs) enable (i) parallel decoding and (ii) any-order decoding. Recently, there has been a rapidly growing interest in dLLMs for their potential to dramatically accelerate LLM inference via parallel decoding. While early diffusion language models suffered from performance gaps compared to autoregressive models, recent advances (Inception Labs et al., 2025; Song et al., 2025) have shown that dLLMs can achieve comparable generation quality to autoregressive LLMs, potentially emerging as next-generation LLMs.

---

*Equal Contribution. Emails: {kangwj1995, kevin.galim, seunghyukoh}@furiosa.ai.

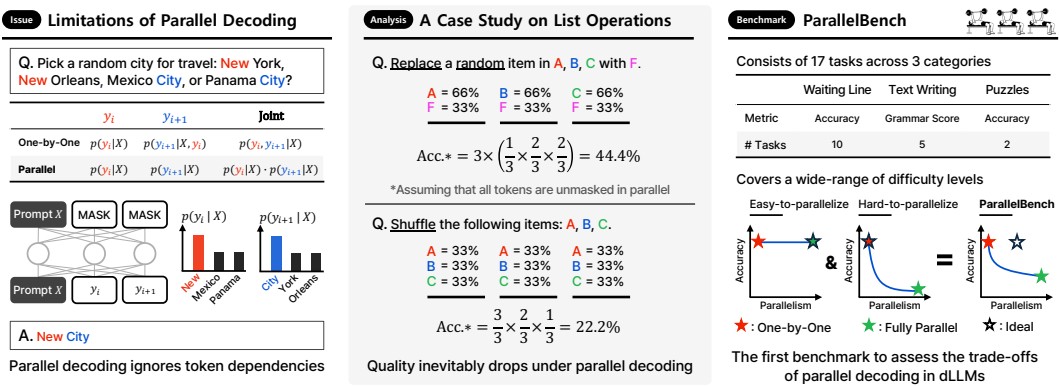

Figure 1: PARALLELBENCH. **(left)** Parallel decoding fails to capture token dependencies, risking incorrect combinations of individually likely tokens (e.g., *"New"* and *"City"*). **(middle)** Quantitative analysis of parallel decoding performance on list operations. **(right)** Based on our findings, we develop a realistic benchmark to evaluate the speed-quality trade-off of parallel decoding methods.

However, the conditional independence assumption (Wu et al., 2025) in dLLMs causes parallel decoding to fail to capture token dependencies, resulting in significant quality degradation. As shown in Fig. 1, under parallel decoding, the model might generate *"New City"* instead of the correct *"New York"* or *"Mexico City"* because each token is unmasked independently without considering its mutual constraints. Existing works on dLLMs (Nie et al., 2025; Ye et al., 2025b) mainly explore their one-by-one decoding quality with limited analysis of parallel decoding challenges, despite parallel generation being their defining advantage. Moreover, current evaluations rely on standard benchmarks, such as math (Cobbe et al., 2021) and coding (Chen et al., 2021), which may not adequately expose the quality degradation of parallel decoding in real-world scenarios.

To address this gap, we first provide an information-theoretic analysis of parallel decoding in dLLMs, showing that even ideal models suffer increasing difficulty due to inherent token dependencies in the data distribution as the degree of parallelism increases. To further provide quantitative insights, we conduct case studies on analytically tractable synthetic list operations from two perspectives: (i) from the data distribution perspective, we quantitatively formulate the difficulty of parallel decoding through conditional total correlation, proving that certain list operations exhibit inevitable quality degradation under parallel decoding, and (ii) from the decoding order perspective, we derive the achievable accuracy of various unmasking strategies on these tasks, showing that accuracy varies significantly across different unmasking strategies.

Building on this, we propose PARALLELBENCH to demonstrate that the quality degradation from parallel decoding extends beyond synthetic to real-world scenarios. This realistic benchmark encompasses tasks of varying difficulty under parallel decoding, including those that are exceptionally challenging for parallel decoding in dLLMs yet still trivial for humans and autoregressive LLMs. Using this benchmark, we conduct extensive evaluations of dLLMs and autoregressive LLMs, revealing two key findings. First, consistent with our synthetic results, dLLMs with parallel decoding exhibit severe quality degradation, even on trivial real-world tasks. Second, current parallel decoding strategies struggle to adaptively adjust the degree of parallelism based on task difficulty, resulting in suboptimal speed-quality trade-offs. Finally, we explore additional techniques to determine if they can improve the current trade-offs in parallel decoding. In summary, our main contributions are:

- We provide an information-theoretic analysis to formalize the quality degradation in parallel decoding arising from token dependencies and design tractable synthetic tasks to quantify its impact.

- We introduce PARALLELBENCH, the first realistic benchmark for evaluating trade-offs of parallel decoding in dLLMs, covering diverse difficulty levels to assess adaptive parallelism strategies.

- Through extensive experiments on PARALLELBENCH, we demonstrate that current dLLMs under parallel decoding suffer severe quality degradation on seemingly simple tasks, and existing decoding strategies struggle to adaptively balance speed and quality.

## 2    RELATED WORKS

**Diffusion-based Language Models**   D3PM (Austin et al., 2021) pioneered applying diffusion models to discrete data by modeling discrete diffusion processes through various transition matrices. Diffusion-LM (Li et al., 2022) introduced a non-autoregressive language model using continuous diffusion. Lou et al. (2024) introduced SEDD, which parameterizes discrete diffusion processes by learning concrete scores through a novel score entropy loss, thereby achieving performance comparable to that of autoregressive models. Sahoo et al. (2024); Shi et al. (2024) showed that masked diffusion with absorbing state achieves stronger performance through a simplified framework. Recent diffusion-based language models (Kim et al., 2025c) have achieved significant improvements in scale and performance, establishing practical dLLMs (Nie et al., 2025; Zhu et al., 2025; Ye et al., 2025b; Kim et al., 2025d) with expanded multimodal capabilities (You et al., 2025; Yang et al., 2025b) and variants for complex reasoning (Zhao et al., 2025) and code generation (Gong et al., 2025; Inception Labs et al., 2025; Song et al., 2025).

**Analysis of Decoding Mechanisms in dLLMs**   Wu et al. (2025) discussed the quality degradation in dLLMs' parallel decoding process due to the conditional independence assumption, proposing threshold-based and factor-based unmasking methods to mitigate this issue. However, they only focus on standard benchmarks like GSM8K (Cobbe et al., 2021) and HumanEval (Chen et al., 2021) rather than exploring settings where parallel decoding is particularly vulnerable. Feng et al. (2025) demonstrated that dLLMs' parallel decoding is highly task-dependent: efficient for fluent text generation with constant steps, but ineffective for reasoning tasks requiring linearly scaling steps for sequence-level correctness. However, this work offers limited validation, relying on highly synthetic settings like n-grams and Hidden Markov Models. Gong et al. (2025) introduced AR-ness metrics to quantify the similarity between dLLM and autoregressive LLM decoding orders. They found that dLLMs exhibit any-order decoding, with AR-ness varying by dataset and correlating with quality degradation in parallel decoding. However, their analysis is limited to standard benchmarks, such as code and math. Ye et al. (2025a) demonstrated that dLLMs surpass autoregressive LLMs in complex reasoning and long-term planning tasks, such as Countdown and Sudoku.

For a broader overview of related work, see Appendix B.

## 3    PRELIMINARIES: DIFFUSION LLM DECODING

Since most dLLMs (Nie et al., 2025; Ye et al., 2025b) employ masked diffusion (Sahoo et al., 2024), we assume masked diffusion throughout this paper. For a more detailed discussion, see Appendix B.3.

**Parallel Decoding**   In parallel decoding, dLLMs generate tokens over $T$ timesteps with target sequence $Y = S_1 \cup S_2 \cup \cdots \cup S_T$. At timestep $t$[1], the model generates token set $S_t$ conditioned on input $X$ and previously generated tokens $S_{<t} = S_1 \cup \ldots \cup S_{t-1}$. The generation probability is factorized as $P_\theta(S_t|X, S_{<t}) = \prod_{y_i \in S_t} P_\theta(y_i|X, S_{<t})$, assuming conditional independence among tokens in $S_t$. This **conditional independence assumption** enables parallel decoding within each timestep but introduces factorization errors from unmodeled token dependencies, creating a fundamental speed-quality trade-off. These errors arise with strong semantic or syntactic dependencies, as shown in Fig. 1: dLLMs sample from factorized marginals $P_\theta(y_i|X) \cdot P_\theta(y_{i+1}|X)$ instead of the true joint $P_\theta(y_i, y_{i+1}|X)$, potentially generating invalid combinations like *"New City"* rather than *"New York"* or *"Mexico City"*. In the special case of **one-step generation** ($T = 1$), Huang et al. (2022) proved that the minimum achievable KL divergence between the factorized model $P_\theta(Y|X) = \prod_{y_i \in Y} P_\theta(y_i|X)$ and the true data distribution $P_{\text{data}}(Y|X)$ is lower-bounded by the conditional total correlation $\mathcal{C}(Y|X) = -H_{\text{data}}(Y|X) + \sum_{y_i \in Y} H_{\text{data}}(y_i|X)$:

$$\min_\theta \mathcal{D}_{\text{KL}}(P_{\text{data}}(Y|X) \parallel P_\theta(Y|X)) \geq \mathcal{C}(Y|X) \tag{1}$$

$\mathcal{C}(Y|X)$ quantifies the difficulty of parallel generation by measuring token dependencies, imposing fundamental limits that even optimally designed models cannot overcome.

---

[1]For intuitive understanding, we use notation $t = 0$ to $t = T$ where $t$ corresponds to the $t$-th parallel decoding step, rather than the typical reverse time notation in diffusion models.

**Any-order Decoding: Token Unmasking Methods** dLLMs can decode in any order, where the unmasking strategy determines how to partition $Y$ into disjoint sets $S_1 \cup \cdots \cup S_T$, critically impacting output quality. At each timestep, dLLMs make predictions for all masked token positions but only unmask and finalize a subset of them, keeping the rest masked for subsequent timesteps. Unmasking methods can be broadly categorized into two approaches: (i) **Top-k (static)**: which unmasks a fixed number of tokens in left-to-right order, randomly, or based on a scoring metric (e.g., confidence (Nie et al., 2025), margin (Kim et al., 2025b), and entropy (Ye et al., 2025b)), and (ii) **Threshold (adaptive)**: which unmasks tokens whose scores exceed a certain threshold; if no tokens meet the threshold, the single token with the highest score is unmasked. Furthermore, several works (Nie et al., 2025; Arriola et al., 2025) adopt semi-autoregressive (semi-AR) decoding, which divides sequences into fixed-size blocks decoded left-to-right, where tokens within each block are generated in parallel using any unmasking strategy once preceding blocks are complete.

## 4 THEORETICAL ANALYSIS OF PARALLEL DECODING

In this section, we theoretically explore the specific quality degradation that occurs in the case of $T$-step parallel decoding, and then provide quantitative analyses from both data distribution and decoding strategy perspectives, using analytically tractable synthetic list operations.

**Theorem 1** (Lower Bound for $T$-step Parallel Decoding). *For a factorized generative model (e.g., dLLM) $P_\theta(Y|X)$ performing T-step parallel decoding, assume the target sequence $Y$ is partitioned into T disjoint sets, $Y = S_1 \cup S_2 \cup \cdots \cup S_T$. At each step $i \in \{1, \ldots, T\}$, the model generates the tokens in set $S_i$ in parallel, conditioned on $X$ and all previously generated tokens $S_{<i} = S_1 \cup S_2 \cup \ldots \cup S_{i-1}$ ($S_{<1} = \emptyset$). The minimum achievable KL divergence for this model is lower-bounded by:*

$$\min_\theta \mathcal{D}_{KL}(P_{data}(Y|X) \parallel P_\theta(Y|X)) \geq \mathcal{L}_T\left(\{S_i\}_{i=1}^T\right) := \sum_{i=1}^T \mathbb{E}_{S_{<i} \sim P_{data}}[\mathcal{C}(S_i|X, S_{<i})] \quad (2)$$

*where $\mathcal{C}(S_i|X, S_{<i})$ denotes the conditional total correlation of tokens in $S_i$ given $X$ and $S_{<i}$. The equality holds when $P_\theta(y_j|X, S_{<i}) = P_{data}(y_j|X, S_{<i})$ for all $i \in \{1, \ldots, T\}$ and for all $y_j \in S_i$.*

**Remark 1** (Boundary Cases). *The lower bound $\mathcal{L}_T$ captures the two boundary cases of the decoding spectrum: i) When $T = 1$, $\mathcal{L}_1 = \mathcal{C}(Y|X)$; ii) When $T = |Y|$ (i.e., one-by-one decoding with $S_i = \{y_i\}$ for all i), for each i we have $\mathcal{C}(S_i|X, S_{<i}) = 0$, thus $\mathcal{L}_{|Y|} = 0$.*

**Theorem 2** (Monotonicity of Error Bounds). *Let $\mathcal{L}_T^*$ denote the optimal (minimum) error bound over all possible T-step partitions $\{S_i\}_{i=1}^T$: $\mathcal{L}_T^* := \min_{\{S_i\}_{i=1}^T} \mathcal{L}_T(\{S_i\}_{i=1}^T)$. Then $\mathcal{L}_T^*$ is monotonically decreasing with respect to the number of generation steps: $\mathcal{L}_T^* \geq \mathcal{L}_{T+1}^*$.*

**Remark 2.** *Since $\mathcal{L}_1^* = \mathcal{C}(Y|X)$ and $\mathcal{L}_{|Y|}^* = 0$, the optimal error bound $\mathcal{L}_T^*$ forms a monotonically decreasing function from the total correlation $\mathcal{C}(Y|X)$ to zero as T increases from 1 to $|Y|$.*

See Appendix A for proofs and Appendix A.3 for the extension of Theorem 2 to input-dependent partitions. Theorem 1 shows that $T$-step parallel decoding in dLLMs inevitably incurs distribution error due to data distribution properties such as conditional total correlation (e.g., $\mathcal{C}(Y|X)$), even with ideal models. Theorem 2 further establishes that the lower bound on this distribution error increases monotonically as $T$ decreases. However, $\mathcal{C}(Y|X)$ is intractable in real-world datasets, though Huang et al. (2022) approximated it with a well-trained transformer.

To address this gap, we use synthetic list operations with analytically tractable $\mathcal{C}(Y|X)$. We examine four tasks on length-$n$ input sequences (e.g., [A, B, C, D, E]): (i) **Copy**: copying the input; (ii) **Replace Index**: replacing an item at a given index with F; (iii) **Replace Random**: replacing one random item with F; and (iv) **Shuffle**: rearranging the items.

### 4.1 A CASE STUDY ON LIST OPERATIONS: A DATA DISTRIBUTION PERSPECTIVE

**Copy & Replace Index** While *Replace Index* seems harder than *Copy*, both tasks are equally simple from the perspective of $\mathcal{C}(Y|X)$. For both tasks, each output token $y_i$ is uniquely determined by $X$ alone ($\mathcal{C}(Y|X) = 0$), allowing ideal models to decode in parallel without distribution error.

**Replace Random** While *Replace Index* seems harder than *Replace Random* (requiring replacement at a given index), from the perspective of $\mathcal{C}(Y|X)$, the opposite holds. While *Replace Index*

yields $\mathcal{C}(Y|X) = 0$, *Replace Random* exhibits token dependencies since replacing exactly one randomly selected item requires all others to remain unchanged, yielding:

$$\mathcal{C}(Y|X) = (n-1)[\log_2(n) - \log_2(n-1)], \quad \lim_{n\to\infty} \mathcal{C}(Y|X) = \log_2(e) \approx 1.44. \quad (3)$$

This indicates that the difficulty of parallel decoding remains bounded for arbitrarily long sequences.

**Shuffle** *Shuffle* generates a random permutation and exhibits stronger token dependencies than *Replace Random*, as once an item is placed at any position, it cannot appear elsewhere. This yields:

$$\mathcal{C}(Y|X) = n\log_2(n) - \log_2(n!), \quad \lim_{n\to\infty} \mathcal{C}(Y|X) = \infty, \quad (4)$$

indicating that the difficulty of capturing permutation constraints grows without bound for arbitrarily long sequences in one-step generation. For $T$-step parallel decoding, $\mathcal{L}_T(\{S_t\}_{t=1}^T) = \sum_{t=1}^T |S_t|\log_2(k_t) - \log_2(n!)$, where $k_t = n - \sum_{j=1}^{t-1} |S_j|$ with $k_1 = n$. When $T = n/2$ (2 tokens per step), $\mathcal{L}_{n/2} = \log_2 \frac{n!!}{(n-1)!!}$, $\lim_{n\to\infty} \mathcal{L}_{n/2} = \infty$, where $n!!$ is the double factorial. This indicates that the difficulty grows without bound even when decoding only 2 tokens in parallel.

## 4.2 A CASE STUDY ON LIST OPERATIONS: A DECODING STRATEGY PERSPECTIVE

Beyond the data distribution perspective, the practical generation quality of dLLMs critically depends on the decoding (unmasking) strategy. We analyze achievable accuracy under two categories: (i) **Top-k** (Random or Confidence): unmask $k$ random positions or $k$ highest-confidence positions per step; (ii) **Threshold** (Confidence): unmask all positions exceeding confidence threshold $\gamma$; if no positions meet $\gamma$, the single token with the highest confidence is unmasked. We focus on an unbiased (ideal) model where each token's logit $l_j = l_{j,\text{ideal}} + \epsilon_j$, with $l_{j,\text{ideal}}$ determined solely by the task and $\epsilon_j$ representing zero-mean stochastic noise, without intrinsic bias. We consider greedy decoding ($\tau = 0$) and temperature sampling ($\tau = 1$). Summary of our findings are in Table 1.

### 4.2.1 TOP-K (RANDOM OR CONFIDENCE)

**Shuffle** Since at each step $i$, the probability of correctly selecting $k$ distinct items for the remaining positions is $\frac{P(n-(i-1)k,k)}{(n-(i-1)k)^k}$, the overall accuracy of successfully completing the shuffle operation is:

$$\text{Acc}(k) = \prod_{i=1}^{n/k} \frac{P(n-(i-1)k,k)}{(n-(i-1)k)^k} = \begin{cases} \frac{n!}{n^n} \to 0 & \text{if } k = n \\ \frac{(n-1)!!}{n!!} \to 0 & \text{if } k = 2 \quad \text{as } n \to \infty, \\ 1 \to 1 & \text{if } k = 1 \end{cases} \quad (5)$$

regardless of using greedy decoding or temperature sampling. Notably, for both $k = n$ (one-step generation) and $k = 2$ (two tokens per step), the accuracy converges to zero as $n$ increases.

**Replace Random** Under greedy decoding (assume $k \mid n$ for simplicity), at each step $i < \frac{n}{k}$, all $k$ positions keep their tokens since the keep probability $\frac{n-(i-1)k-1}{n-(i-1)k}$ exceeds the replace probability $\frac{1}{n-(i-1)k}$. At the final step, when $k > 2$, all positions are kept since $\frac{k-1}{k} > \frac{1}{k}$, failing the task; when $k = 2$, a successful replacement occurs with probability $0.5 \times 0.5 + 0.5 \times 0.5 = 0.5$ (sum of [keep, replace] and [replace, keep]). Thus $\text{Acc}(k > 2) = 0$ and $\text{Acc}(k = 2) = 0.5$.

Under temperature sampling, each position initially keeps its original token with probability $\frac{n-1}{n}$. Thus, for one-step generation, we have $\text{Acc}(n) = \left(\frac{n-1}{n}\right)^{n-1}$, and $\lim_{n\to\infty} \text{Acc}(n) = \frac{1}{e}$.

### 4.2.2 THRESHOLD (CONFIDENCE)

**Shuffle** Assuming $\gamma > 0.5$, at each timestep, the confidence for each item at each position equals $1/m$ where $m$ is the number of remaining masked tokens. For $m \geq 2$, we have $\gamma > 0.5 \geq 1/m$, so the threshold is never met when multiple tokens remain masked. Thus, only a single token is decoded at each timestep, converging to one-by-one decoding with guaranteed success: $\text{Acc}(\gamma > 0.5) = 1$.

**Replace Random**  Under greedy decoding with $n > 2$, each position at the first timestep assigns a confidence of $\frac{n-1}{n}$ to retaining its original token and $\frac{1}{n}$ to replacement, thus all positions greedily preserve their original tokens. When $\frac{n-1}{n} > \gamma$, all positions are unmasked simultaneously, resulting in one-step generation that fails to perform any replacements: $\text{Acc}(\frac{n-1}{n} > \gamma) = 0$. When $\gamma \geq \frac{n-1}{n}$, all positions fail to exceed the threshold, so only one token is unmasked per timestep. This one-by-one decoding continues through subsequent timesteps until the final step, where only a single token remains masked. At this final step, the model assigns confidence 1 to the correct replacement, ensuring successful completion: $\text{Acc}(\gamma \geq \frac{n-1}{n}) = 1$.

Table 1: Summary of findings in Section 4.

| Task / Analysis | Data Distribution Perspective (Section 4.1) | | Decoding Strategy Perspective (Section 4.2) | |
| --- | --- | --- | --- | --- |
| | $\mathcal{C}(Y|X)$ | $\lim_{n\to\infty}\mathcal{C}(Y|X)$ | Acc. (Greedy, Top-k) | Acc. (Greedy, Threshold) |
| **Copy & Replace Index** | 0 | 0 | 1 | 1 |
| **Replace Random** | $(n-1)[\log_2(n) - \log_2(n-1)]$ | $\log_2(e) \approx 1.44$ | 0.5 if $k=2$; 0 if $k>2$ | 1 if $\gamma \geq (n-1)/n$; else 0 |
| **Shuffle** | $n\log_2(n) - \log_2(n!)$ | $\infty$ | Eq. (5) | 1 if $\gamma > 0.5$ |

### 4.2.3 EMPIRICAL VALIDATION

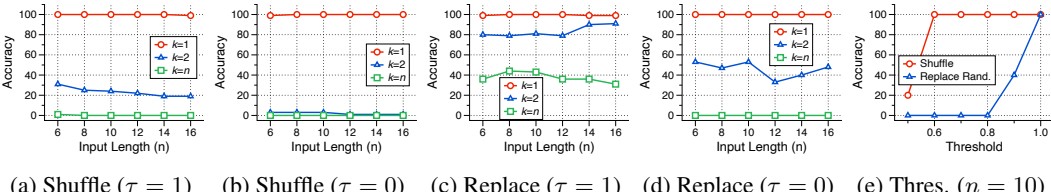

(a) Shuffle ($\tau = 1$)  (b) Shuffle ($\tau = 0$)  (c) Replace ($\tau = 1$)  (d) Replace ($\tau = 0$)  (e) Thres. ($n = 10$)

Figure 2: Empirical validation results on list operations: *Shuffle* (Figs. 2a and 2b) and *Replace Random* (Figs. 2c and 2d) with *Random Top-k*, and both tasks with *Confidence Threshold* (Fig. 2e).

To validate our analysis, we fine-tune LLaDA 1.5 (Zhu et al., 2025) on each list operation task and perform infilling experiments with pre-filled formatting tokens, leaving only the list item positions to be generated. Figs. 2a and 2b validate Eq. (5): accuracy converges to 0 as $n$ increases for $k > 1$, with $k = n$ converging much faster than $k = 2$. Fig. 2c confirms that one-step generation achieves $1/e$ accuracy under temperature sampling, while Fig. 2d shows that greedy decoding yields 0.5 accuracy for $k = 2$ but 0 for $k = n$. Fig. 2e confirms that *Shuffle* and *Replace Random* exhibit opposite trends with varying $\gamma$, indicating the difficulty of achieving high accuracy for both tasks.

## 5 REALISTIC BENCHMARK: PARALLELBENCH

While we reveal limitations of parallel decoding in synthetic settings, we demonstrate that these issues persist in real-world scenarios by introducing PARALLELBENCH, a realistic benchmark spanning diverse difficulty levels to evaluate dLLMs' parallel decoding. It consists of 17 tasks across 3 categories: (i) **Waiting Line** (10 tasks), (ii) **Text Writing** (5 tasks), and (iii) **Puzzles** (2 tasks).

**Waiting Line**  *Waiting Line* extends synthetic list operations (e.g., *Shuffle*) to realistic customer service scenarios. Given a queue of customers (e.g., ["Susan Fox", "Philip Gray", "Maria Butler", "Albert Sanchez"]), the task performs queue management operations analogous to list operations. The number of customers $n$ serves as a controllable parameter for adjusting benchmark difficulty.

**Text Writing**  To expose the quality degradation in parallel decoding, we introduce grammar (Morris, 2025) scores that impose stronger token-level dependencies, which traditional metrics such as ROUGE (Lin, 2004) cannot capture. We examine five tasks: (i) *Summarization* using SAMSum (Gliwa et al., 2019), (ii) *Paraphrasing* using Vorobev & Kuznetsov (2023), and (iii-v) our proposed *Words-to-Sentence Generation (W2S)* with three difficulty levels. Both *Summarization* and *Paraphrasing* constrain outputs to rich input context, limiting token candidates and keeping $\mathcal{C}(Y|X)$ small. For scenarios with greater token dependencies, we propose *W2S*, where the goal is to construct a sentence using the given $n$ words, such as *"Construct a single, coherent sentence using the*

*words sand, home, play, and bottle."* Since the output can be freely generated using minimal input context, the token dependencies become larger depending on what output is generated. We design three difficulty levels: *easy* (related words), *medium* (loosely connected words), and *hard* (unrelated words), with harder variants requiring more creative construction and exhibiting larger $\mathcal{C}(Y|X)$.

**Puzzles** Ye et al. (2025a;b) utilized *Sudoku* to demonstrate the superior planning capabilities of dLLMs over autoregressive LLMs. Crucially, every Sudoku puzzle, regardless of difficulty, has a unique solution ($\mathcal{C}(Y|X) = 0$). While *Latin Square* (Keedwell & Dénes, 2015) shares structural similarities with *Sudoku*, it has many valid solutions ($\mathcal{C}(Y|X) > 0$). We included both to examine how $\mathcal{C}(Y|X)$ affects parallel decoding in structurally similar tasks.

## 6 BENCHMARK RESULTS AND ANALYSIS ON PARALLELBENCH

**Setup** For autoregressive LLMs, we use Llama 3.1 8B, Llama 3.2 3B (Grattafiori et al., 2024), Qwen2.5 3B/7B (Yang et al., 2024), Qwen3 4B (Yang et al., 2025a), and Claude 3.5 Haiku (Anthropic, 2024). For dLLMs, we test LLaDA 8B (Nie et al., 2025), LLaDA 1.5 8B (Zhu et al., 2025), Dream 7B (Ye et al., 2025b), DiffuCoder (Gong et al., 2025), and closed-source Mercury (Inception Labs et al., 2025). We also evaluate KV caching (Wu et al., 2025). For unmasking methods, we test: i) *Top-k*: Random, Confidence (Chang et al., 2022), Left-to-Right[2], Margin (Kim et al., 2025b), Entropy (Ye et al., 2025b); and ii) *Threshold*: Confidence (Wu et al., 2025) with $\gamma \in \{0.5, 0.6, \dots, 1.0\}$. We also test *Factor-based* (Wu et al., 2025), and semi-AR decoding (Nie et al., 2025). Further experimental details are in Appendix C. In this section, we primarily focus on LLaDA 1.5, a representative open-source dLLM, and four unmasking methods with results shown in Fig. 3, while full analysis including additional models, unmasking methods, and the impact of KV caching is in Appendix D. When evaluating LLaDA 1.5 on *Waiting Line*, we use $n \in \{3, 4, 5, 6\}$, ensuring task simplicity to focus on parallel decoding effects rather than model capacity limitations.

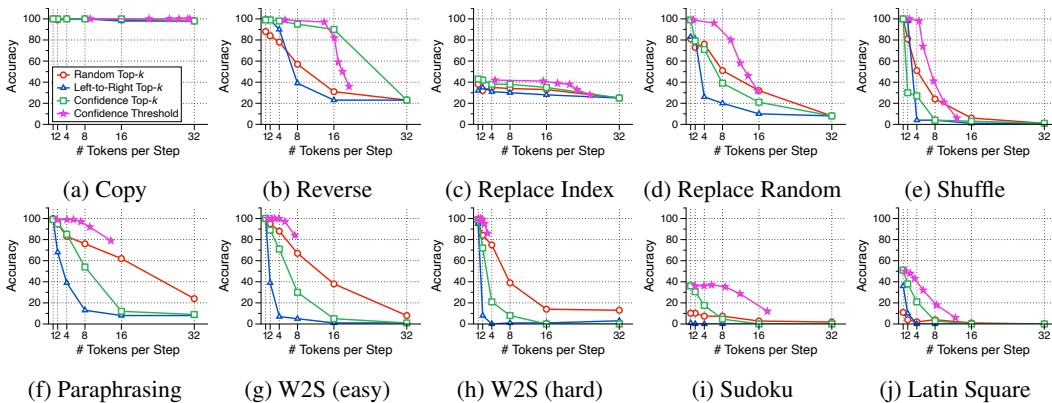

Figure 3: Benchmark results of LLaDA 1.5 (Zhu et al., 2025) on PARALLELBENCH: *Waiting Line* (Figs. 3a to 3e), *Text Writing* (Figs. 3f to 3h), and *Puzzles* (Figs. 3i and 3j).

**Comparison Across Tasks** For *Waiting Line*, *Copy* maintains near-perfect accuracy regardless of parallelism degree (Fig. 3a). For *Reverse* (Fig. 3b), accuracy varies by unmasking method. We argue that these differences stem from insufficient model capacity rather than token dependency ($\mathcal{C}(Y|X) = 0$). *Replace Index* (Fig. 3c) and *Replace Random* (Fig. 3d) exhibit contrasting trends. Despite *Replace Random* achieving near-perfect one-by-one decoding accuracy compared to *Replace Index*'s below 50% accuracy, this advantage reverses under parallelization: *Replace Index* maintains stable accuracy with increasing parallelism, while *Replace Random* degrades rapidly across all unmasking methods. This reversal aligns with Section 4: $\mathcal{C}(Y|X) = 0$ for *Replace Index* but $\mathcal{C}(Y|X) > 0$ for *Replace Random*. *Shuffle* (Fig. 3e) shows the steepest accuracy degradation, dropping from near-perfect to zero more rapidly than *Replace Random* as parallelism increases. For

---

[2]Unmasks $k$ tokens at a time from left to right.

***Text Writing*** (Figs. 3f to 3h), all tasks achieve near-perfect grammar scores with one-by-one decoding, but degrade at increasingly steep rates under parallelization: *Paraphrasing*, *W2S (easy)*, then *W2S (hard)*. This aligns with expected conditional dependencies. *Paraphrasing* merely restructures existing content with rich context, limiting token candidates and keeping $\mathcal{C}(Y|X)$ small. Conversely, *W2S* tasks generate complete sentences from sparse word constraints, expanding the candidate space where each position strongly influences others. *W2S (hard)* uses semantically unrelated words requiring more creative construction than *W2S (easy)*'s naturally coherent word sets, resulting in larger $\mathcal{C}(Y|X)$ and steeper degradation. For ***Puzzles*** (Figs. 3i and 3j), *Latin Square* outperforms the harder *Sudoku* under one-by-one decoding, but this advantage disappears with increased parallelism (*Latin Square*: $\mathcal{C}(Y|X) > 0$, *Sudoku*: $\mathcal{C}(Y|X) = 0$), leading to similar accuracy.

**Comparison Across Unmasking Methods**   *Confidence Threshold* outperforms *Top-k* at conservative thresholds through adaptive token selection, but suffers rapid degradation at aggressive thresholds due to unpredictable unmasking spikes, making careful tuning essential. When comparing among top-k methods, for *Reverse* and *Replace Index*, *Confidence Top-k* outperforms *Random Top-k*. This is because when $\mathcal{C}(Y|X) = 0$, model imperfection limits accuracy, making it beneficial to unmask tokens that the model is confident about first. For *Replace Random* and *Shuffle* ($\mathcal{C}(Y|X) > 0$), *Random Top-k* conversely outperforms *Confidence Top-k*. In conclusion, no universally superior unmasking method exists, with performance varying by task.

**Semi-AR Decoding Results**   Semi-AR decoding controls the left-to-right tendency in unmasking order by adjusting block length. Its effectiveness in parallel decoding depends on the dependency structure. With local dependencies (e.g., grammatical constraints in *Text Writing*), small blocks enforce left-to-right (local) decoding, causing quality degradation. With global dependencies (e.g., *Waiting Line*'s distributed items), left-to-right decoding becomes beneficial by capturing formatting tokens around items. Fig. 5 shows different accuracy trends between *Text Writing* and *Waiting Line* across block sizes (*Random Top-2*), indicating the difficulty of optimizing both tasks simultaneously.

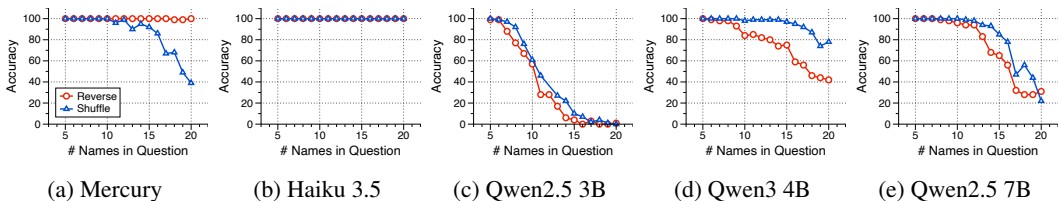

(a) Mercury          (b) Haiku 3.5          (c) Qwen2.5 3B          (d) Qwen3 4B          (e) Qwen2.5 7B

Figure 4: *Waiting Line* results on Mercury (Inception Labs et al., 2025) and autoregressive LLMs.

**Performance of Mercury vs. Autoregressive LLMs**   Fig. 4 shows *Waiting Line* results with $n \in [5, 20]$, where Mercury and autoregressive LLMs show opposite accuracy trends on *Reverse* and *Shuffle*. As *Reverse* seems harder (requiring exact reversal) than *Shuffle* (accepting any permutation), autoregressive LLMs score higher on *Shuffle*. However, Mercury shows the opposite: it maintains near-perfect accuracy on *Reverse* but fails with *Shuffle*, with accuracy dropping sharply as $n$ increases, consistent with Section 4. This indicates that the closed-source Mercury struggles to adaptively adjust its parallelism to maintain quality on tasks with high token dependencies.

**Broad Coverage of PARALLELBENCH**   Fig. 6 demonstrates PARALLELBENCH's necessity and differentiation from existing benchmarks. The x-axis shows parallelism-induced quality degradation (rightward = greater), while the y-axis shows semi-AR block length effects (upward = better quality with left-to-right ordering). For *Waiting Line*, tasks with $\mathcal{C}(Y|X) = 0$ appear left, while those with $\mathcal{C}(Y|X) > 0$ progressively move rightward with increasing degradation, culminating in *Shuffle* at the far right. For *Text Writing*, lower dependency tasks (*paraphrasing*, *summarization*) appear left, while higher dependency *W2S* tasks appear right. In contrast to existing benchmarks like GSM8K (Cobbe et al., 2021), MATH (Hendrycks et al., 2021), and IFEval (Zhou et al., 2023), which occupy a narrow range, PARALLELBENCH spans a broad spectrum of parallel decoding difficulty. This broad coverage uniquely enables the evaluation of adaptive methods' ability to modulate parallelism according to task difficulty. Details are in Appendix F.1.

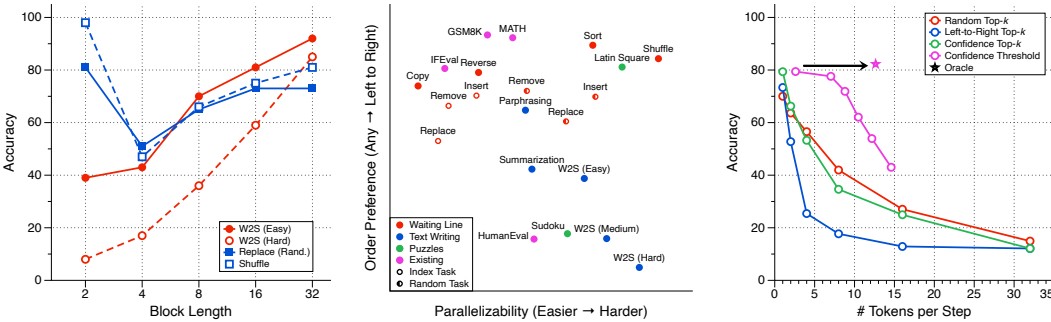

Figure 5: Semi-AR results on *Waiting Line* and *Text Writing*.

Figure 6: Broad coverage of PARALLELBENCH.

Figure 7: Speed-quality trade-off with oracle performance.

**Speed-Quality Trade-off with Oracle Performance**  Fig. 7 shows the speed-quality trade-off curves for each unmasking method. Static unmasking (top-k) methods show dramatic accuracy drops as tokens-per-step increase, while adaptive unmasking (threshold) achieves superior trade-offs. This raises the question: *Do existing adaptive unmasking methods already provide sufficient speedup-quality trade-offs?* To answer this, we measure oracle performance: the best possible trade-off assuming access to the optimal threshold for each sample that yields the correct answer. The oracle achieves both the best accuracy and a significant speedup over threshold methods with comparable accuracy, demonstrating that per-sample threshold adaptation alone yields significant improvements and suggests a substantial room for future research. Details are in Appendix F.2.

> **Takeaway.**  Static parallel decoding (e.g., top-k) can suffer severe quality degradation, and adaptive decoding strategies (e.g., threshold) still have significant room for improvement.

# 7 EXPLORING ADDITIONAL TECHNIQUES

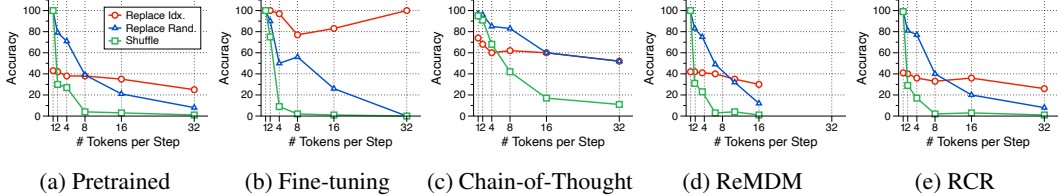

(a) Pretrained    (b) Fine-tuning    (c) Chain-of-Thought    (d) ReMDM    (e) RCR

Figure 8: *Waiting Line* results using LLaDA 1.5 (*Random Top-k*) with various advanced techniques.

We further explore whether benchmark performance can be improved when various advanced techniques for dLLMs are applied. Details and full results are in Appendix E.

**Additional Unmasking Methods**  Beyond the methods evaluated in our main experiments, several recent works (Ben-Hamu et al., 2025; Luxembourg et al., 2025; Wei et al., 2025; Huang et al., 2025) propose advanced unmasking strategies to improve performance in dLLMs. For an overview of these methods, see Appendix B.1. We evaluate their speed-quality trade-offs on PARALLELBENCH and report results in Appendix E.1.

**Fine-tuning**  We fine-tuned LLaDA 1.5 on *Waiting Line* to examine whether fine-tuning improves parallel decoding performance (Figs. 8a and 8b). One-by-one decoding achieves near 100% accuracy for all tasks after fine-tuning, including *Replace Index*, which improves from below 50% to nearly 100%. Notably, *Replace Index* maintains high accuracy even with parallel decoding, confirming that ideal models can accurately learn the target distribution under parallel generation when $C(Y|X) = 0$. However, *Replace Random* and *Shuffle* still degrade with parallel decoding, supporting Section 4 that even an ideal model cannot resolve parallel decoding limitations for $C(Y|X) > 0$. Details and full results are in Appendix E.2.

**Chain-of-Thought Prompting (CoT)**    Figs. 8a and 8c show that CoT mitigates quality degradation under increased parallelism by generating intermediate reasoning steps that reduce inter-token dependencies in final answers. However, due to using $8\times$ more output tokens, CoT cannot serve as a fundamental solution for improving the speed-accuracy trade-off under parallel decoding. Details and full results are provided in Appendix E.3.

**Remasking Samplers**    Most dLLMs use masked diffusion, where tokens unmasked in previous timesteps cannot be refined later. However, this prevents correcting inaccurate predictions from early timesteps, leading to quality degradation. Recent training-free samplers, such as ReMDM (Wang et al., 2025) and RCR (He et al., 2025), enable remasking for masked diffusion models. However, our tests on *Waiting Line* show no improvements, revealing limitations of training-free approaches. More recently, several works (Zhang et al., 2025; Kim et al., 2025a) demonstrate that finetuning dLLMs to support remasking can further improve performance. Details are in Appendix E.4.

**Discrete Diffusion with Uniform Transition Matrix**    Unlike masked diffusion, discrete diffusion with uniform transition matrix enables iterative refinement of all tokens throughout the denoising process, potentially correcting errors from conditional independence assumptions in parallel decoding. We test this by fine-tuning masked and uniform diffusion models from SEDD (Lou et al., 2024). See Appendix E.5 for details and full results.

## 8    Conclusion

This paper investigates the trade-offs of parallel decoding in dLLMs. We provide an information-theoretic analysis and conduct case studies using synthetic list operations, examining both data distribution and decoding strategy perspectives to reveal quantitative insights into the limitations of parallel decoding. Building on these findings, we propose PARALLELBENCH, a new benchmark demonstrating that limitations observed in synthetic scenarios similarly manifest in real-world applications. Using our benchmark, we draw two key conclusions: (1) dLLMs with parallel decoding can suffer severe quality degradation in real-world scenarios, and (2) existing decoding strategies fail to adaptively adjust parallelism based on task difficulty for optimal speed-quality trade-offs.

**Limitations and Future Works**    First, while our benchmark comprises 3 realistic categories and 17 tasks, broader coverage of real-world scenarios would be beneficial. Second, we primarily analyzed short output sequences to focus on fundamental characteristics, and tasks requiring longer sequences may yield different results. While our adjustable-length tasks naturally support longer sequence evaluation and we provide a preliminary analysis in Appendix D.5, a more thorough study remains an important direction for future work. Finally, leveraging PARALLELBENCH to develop novel unmasking methods that address current parallel decoding limitations is an important goal.

## Acknowledgment

Kangwook Lee is supported by NSF Award DMS-2023239, NSF CAREER Award CCF-2339978, Amazon Research Award, and a grant from FuriosaAI.

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

# Appendix

## A  MATHEMATICAL PROOFS

### A.1  PROOF OF THEOREM 1

*Proof.* The $T$-step parallel decoding model is defined by the factorization:

$$P_\theta(Y|X) = \prod_{i=1}^{T} P_\theta(S_i|X, S_{<i}) \tag{6}$$

Similarly, we can decompose the true data distribution according to the partition $\{S_i\}_{i=1}^{T}$:

$$P_{\text{data}}(Y|X) = \prod_{i=1}^{T} P_{\text{data}}(S_i|X, S_{<i}) \tag{7}$$

Substituting these decompositions into the KL divergence formula, we get:

$$\mathcal{D}_{\text{KL}}(P_{\text{data}}(Y|X) \| P_\theta(Y|X))$$
$$= \mathbb{E}_{P_{\text{data}}} \left[ \log \frac{P_{\text{data}}(Y|X)}{P_\theta(Y|X)} \right] \tag{8}$$

$$= \mathbb{E}_{P_{\text{data}}} \left[ \log \frac{\prod_{i=1}^{T} P_{\text{data}}(S_i|X, S_{<i})}{\prod_{i=1}^{T} P_\theta(S_i|X, S_{<i})} \right] \tag{9}$$

$$= \sum_{i=1}^{T} \mathbb{E}_{P_{\text{data}}} \left[ \log \frac{P_{\text{data}}(S_i|X, S_{<i})}{P_\theta(S_i|X, S_{<i})} \right] \tag{10}$$

$$= \sum_{i=1}^{T} \mathbb{E}_{S_{<i} \sim P_{\text{data}}} \left[ \mathcal{D}_{\text{KL}}(P_{\text{data}}(S_i|X, S_{<i}) \| P_\theta(S_i|X, S_{<i})) \right] \tag{11}$$

At each step $i$, the model assumes conditional independence among the tokens in $S_i$ given the context $(X, S_{<i})$, such that $P_\theta(S_i|X, S_{<i}) = \prod_{y_j \in S_i} P_\theta(y_j|X, S_{<i})$. Therefore, by applying Eq. (1) from Section 3, we have:

$$\mathcal{D}_{\text{KL}}(P_{\text{data}}(S_i|X, S_{<i}) \| P_\theta(S_i|X, S_{<i})) \geq \mathcal{C}(S_i|X, S_{<i}) \tag{12}$$

By substituting this lower bound back into our sum, we have:

$$\min_\theta \mathcal{D}_{\text{KL}}(P_{\text{data}}(Y|X) \| P_\theta(Y|X))$$
$$= \min_\theta \sum_{i=1}^{T} \mathbb{E}_{S_{<i} \sim P_{\text{data}}} \left[ \mathcal{D}_{\text{KL}}(P_{\text{data}}(S_i|X, S_{<i}) \| P_\theta(S_i|X, S_{<i})) \right] \tag{13}$$

$$\geq \sum_{i=1}^{T} \mathbb{E}_{S_{<i} \sim P_{\text{data}}} \left[ \min_{\theta_i} \mathcal{D}_{\text{KL}}(P_{\text{data}}(S_i|X, S_{<i}) \| P_\theta(S_i|X, S_{<i})) \right] \tag{14}$$

$$\geq \sum_{i=1}^{T} \mathbb{E}_{S_{<i} \sim P_{\text{data}}} [\mathcal{C}(S_i|X, S_{<i})] \tag{15}$$

This completes the proof. $\qquad\square$

### A.2  PROOF OF THEOREM 2

**Lemma 1** (Subadditivity of Conditional Total Correlation). *Let a set of random variables $S$ be partitioned into two disjoint subsets $A$ and $B$ (i.e., $S = A \cup B$). For any conditioning context $Z$, the conditional total correlation of $S$ is lower-bounded as follows:*

$$\mathcal{C}(S|Z) \geq \mathcal{C}(A|Z) + \mathbb{E}_{A \sim P_{data}(A|Z)}[\mathcal{C}(B|Z, A)] \tag{16}$$

*Proof of Lemma 1.* Using the definition of conditional total correlation ($\mathcal{C}(S|Z) = \sum_{y \in S} H(y|Z) - H(S|Z)$), we have:

$$\mathcal{C}(A|Z) + \mathbb{E}_A[\mathcal{C}(B|Z,A)]$$

$$= \sum_{y_a \in A} H(y_a|Z) - H(A|Z) + \sum_{y_b \in B} H(y_b|Z,A) - H(B|Z,A) \tag{17}$$

$$\leq \sum_{y_a \in A} H(y_a|Z) - H(A|Z) + \sum_{y_b \in B} H(y_b|Z) - H(B|Z,A) \quad \text{(since } H(y_b|Z) \geq H(y_b|Z,A)\text{)}$$

$$\tag{18}$$

$$= \left( \sum_{y_a \in A} H(y_a|Z) + \sum_{y_b \in B} H(y_b|Z) \right) - (H(A|Z) + H(B|Z,A)) \tag{19}$$

$$= \sum_{y \in S} H(y|Z) - H(A,B|Z) \quad \text{(by the chain rule of entropy)} \tag{20}$$

$$= \mathcal{C}(S|Z) \tag{21}$$

This completes the proof. $\square$

*Proof of Theorem 2.* Let $\{S_i^*\}_{i=1}^T$ be an optimal $T$-step partition that achieves the minimum error bound $\mathcal{L}_T^*$. We construct a $(T+1)$-step partition $\{S_j'\}_{j=1}^{T+1}$ by splitting the last set of the optimal partition, $S_T^*$, into two non-empty, disjoint subsets $A$ and $B$ (i.e., $S_T^* = A \cup B$). The new partition is $\{S_1^*, \dots, S_{T-1}^*, A, B\}$, and we have:

$$\mathcal{L}_T^* = \mathcal{L}_T(\{S_i^*\}_{i=1}^T) \tag{22}$$

$$= \sum_{i=1}^{T-1} \mathbb{E}[\mathcal{C}(S_i^*|X, S_{<i}^*)] + \mathbb{E}[\mathcal{C}(S_T^*|X, S_{<T}^*)] \tag{23}$$

$$\geq \sum_{i=1}^{T-1} \mathbb{E}[\mathcal{C}(S_i^*|X, S_{<i}^*)] + \mathbb{E}[\mathcal{C}(A|X, S_{<T}^*)] + \mathbb{E}[\mathcal{C}(B|X, S_{<T}^*, A)] \quad \text{(by Lemma 1)} \tag{24}$$

$$= \mathcal{L}_{T+1}(\{S_j'\}_{j=1}^{T+1}) \quad \text{(by Theorem 1)} \tag{25}$$

$$\geq \min_{\{S_k\}_{k=1}^{T+1}} \mathcal{L}_{T+1}(\{S_k\}_{k=1}^{T+1}) \tag{26}$$

$$= \mathcal{L}_{T+1}^* \tag{27}$$

Thus, we have $\mathcal{L}_T^* \geq \mathcal{L}_{T+1}^*$. $\square$

## A.3 EXTENDED VERSION OF THEOREM 2

**Theorem 3** (Monotonicity of Error Bounds for Input-Dependent Partitions). *For any given input $X$, let $\mathcal{L}_T(\{S_i\}_{i=1}^T|X)$ be the error bound for a specific $T$-step partition $\{S_i\}_{i=1}^T$:*

$$\mathcal{L}_T(\{S_i\}_{i=1}^T|X) := \sum_{i=1}^T \mathbb{E}_{S_{<i} \sim P_{data}(\cdot|X)}[\mathcal{C}(S_i|X, S_{<i})] \tag{28}$$

*The optimal $T$-step error bound, $\mathcal{L}_T^*$, is the expected value of the minimum error bound over all possible partitions for each input:*

$$\mathcal{L}_T^* := \mathbb{E}_{X \sim P_{data}} \left[ \min_{\{S_i\}_{i=1}^T} \mathcal{L}_T(\{S_i\}_{i=1}^T|X) \right]$$

*Then $\mathcal{L}_T^*$ is monotonically decreasing with respect to the number of generation steps:*

$$\mathcal{L}_T^* \geq \mathcal{L}_{T+1}^* \tag{29}$$

*Proof of Theorem 3.* We begin by establishing a pointwise inequality for any single, arbitrary input $x$. Let us consider a special data distribution concentrated entirely on this single input $x$.

Applying the conclusion of Theorem 2 to this special case directly establishes the following relationship. The optimal T-step error for this distribution (which is simply the optimal error for the input $x$) is greater than or equal to the optimal (T+1)-step error:

$$\min_{\{S_i\}_{i=1}^T} \mathcal{L}_T(\{S_i\}_{i=1}^T | x) \geq \min_{\{S_j'\}_{j=1}^{T+1}} \mathcal{L}_{T+1}(\{S_j'\}_{j=1}^{T+1} | x) \tag{30}$$

Taking the expectation of this pointwise inequality Eq. (30) over the true data distribution $X \sim P_{\text{data}}$ directly yields the final result:

$$\mathcal{L}_T^* := \mathbb{E}_X \left[ \min_{\{S_i\}_{i=1}^T} \mathcal{L}_T(\{S_i\}_{i=1}^T | X) \right] \geq \mathbb{E}_X \left[ \min_{\{S_j'\}_{j=1}^{T+1}} \mathcal{L}_{T+1}(\{S_j'\}_{j=1}^{T+1} | X) \right] =: \mathcal{L}_{T+1}^*$$

Thus, we have $\mathcal{L}_T^* \geq \mathcal{L}_{T+1}^*$. □

## B  FURTHER DISCUSSION ON RELATED WORK

### B.1  ADDITIONAL UNMASKING METHODS

Recent work on dLLMs has focused on optimizing the unmasking process to improve inference efficiency and generation quality. Besides the unmasking methods studied in this work, several additional methods (Ben-Hamu et al., 2025; Luxembourg et al., 2025; Wei et al., 2025; Huang et al., 2025; Wu & Zhang, 2025; Israel et al., 2025) target acceleration through parallel decoding. For instance, the EB-Sampler (Ben-Hamu et al., 2025) adaptively unmasks the largest group of tokens whose cumulative entropy falls below a threshold, $\gamma$, achieving a 2–3× speedup. Similarly, the Dilated Unmasking Scheduler (DUS) (Luxembourg et al., 2025) uses a deterministic, coarse-to-fine schedule to unmask non-adjacent tokens, reducing the number of denoiser calls within a block of size $B$ from $\mathcal{O}(B)$ to $\mathcal{O}(\log B)$. The SlowFast Sampling (Wei et al., 2025) alternates between slow, exploratory phases and fast, parallel decoding phases, yielding up to a 15.6× speedup. Other works focus on generation quality. The PC-Sampler (Huang et al., 2025), for example, corrects for biases in uncertainty-based sampling by using a composite score that combines a position-aware weight with a frequency-calibrated confidence score, improving performance by over 10%.

### B.2  RELATED BENCHMARKS

To contextualize our work, we review several key benchmarks commonly used to evaluate the capabilities of LLMs across different domains.

**GSM8K**  The Grade School Math 8K (GSM8K) dataset (Cobbe et al., 2021) is a prominent benchmark designed to assess the multi-step mathematical reasoning abilities of LLMs. It consists of thousands of grade-school-level word problems that require a sequence of logical steps and arithmetic operations to solve. The primary evaluation metric is final answer accuracy, where the model's generated solution must exactly match the correct numerical result. Success on GSM8K is considered a strong indicator of a model's capacity for complex, chain-of-thought reasoning.

**HumanEval**  The HumanEval benchmark (Chen et al., 2021) is a standard for evaluating the code generation capabilities of models. The task involves completing Python function bodies based on a given function signature and a descriptive docstring. The dataset contains 164 such programming challenges of varying difficulty. Performance is measured by functional correctness, typically using the pass@$k$ (Chen et al., 2021) metric.

**SAMSum**  The SAMSum dataset (Gliwa et al., 2019) is tailored for the task of abstractive summarization of natural, real-world dialogues. It contains short, informal conversations, requiring a model to produce a concise summary that captures the main points of the exchange. The standard evaluation protocol relies on the ROUGE score (Lin, 2004), which measures the n-gram overlap between the model-generated summary and one or more human-written reference summaries. While ROUGE is effective for assessing content overlap, our work diverges by focusing on the grammatical correctness and fluency of the generated summaries, aspects of quality not fully captured by lexical metrics.

**ChatGPT-Paraphrases**    The ChatGPT-Paraphrases dataset (Vorobev & Kuznetsov, 2023) is designed for evaluating a model's ability to rephrase sentences while preserving their original semantic meaning. The dataset provides pairs of original sentences and their corresponding paraphrases. This benchmark is crucial for assessing a model's fine-grained understanding of language and its capacity for fluent and diverse text generation, which are core skills for sophisticated natural language processing applications. While standard evaluations for this task often rely on semantic similarity metrics, our approach is different. Similar to SAMSum, we are primarily interested in the grammatical evaluation of the output.

### B.3    Discrete Diffusion Strategies

In discrete diffusion, a common strategy is the absorbing state approach, also known as masked diffusion. This method corrupts data by progressively replacing original tokens with a single, special `[MASK]` token. The key advantage of this technique lies in its reverse process. The model's task is simplified to "filling in the blanks" at known `[MASK]` locations, which provides a very direct and clear learning signal. This is the primary approach used by many prominent open-source models, such as LLaDA (Nie et al., 2025) and Dream (Ye et al., 2025b).

In contrast, the uniform transition approach corrupts data differently. Instead of using a special token, it substitutes an existing token with any other token from the vocabulary, with each possibility having an equal probability of occurrence. This method gradually transforms the input into a sequence of purely random tokens. Consequently, the reverse process is more complex: the model must predict the original token from a noisy, corrupted one, rather than from a designated blank slate. Some recent models, including SEDD (Lou et al., 2024), have explored this uniform approach.

## C    Benchmark Dataset Specifications and Evaluation Details

Table 2: Benchmark tasks and evaluation metrics of PARALLELBENCH.

| Category | Task | #Samples | Metric |
|---|---|---|---|
| Waiting Line (one-shot) | Copy | 100 | Accuracy |
| | Sort | 100 | Accuracy |
| | Reverse | 100 | Accuracy |
| | Shuffle | 100 | Accuracy |
| | Replace Index | 100 | Accuracy |
| | Replace Random | 100 | Accuracy |
| | Insert Index | 100 | Accuracy |
| | Insert Random | 100 | Accuracy |
| | Remove Index | 100 | Accuracy |
| | Remove Random | 100 | Accuracy |
| Text Writing (zero-shot) | Summarization | 100 | Grammar, Rouge-L |
| | Paraphrasing | 100 | Grammar, BERTScore, (1 - BLEU) |
| | Words-to-Sentence (easy) | 100 | Grammar, Accuracy |
| | Words-to-Sentence (medium) | 100 | Grammar, Accuracy |
| | Words-to-Sentence (hard) | 100 | Grammar, Accuracy |
| Puzzle (one-shot) | Latin Square (4x4) | 100 | Accuracy |
| | Sudoku (4x4) | 108 | Accuracy |

Table 2 provides a comprehensive overview of PARALLELBENCH. We omit the synthetic list operations from this discussion, as their setup is analogous to the *Waiting Line* tasks. Example prompts and corresponding answers for each task are presented in Table 3.

**Evaluation Details**    For the *Waiting Line* and *Puzzle* tasks, which are relatively unfamiliar to the models, we employ one-shot prompting. In this setting, the model is first presented with a complete example (a prompt and its corresponding solution) before being given the actual test prompt. In contrast, for *Text Writing* tasks, we use standard zero-shot prompting.

All models are evaluated using greedy decoding, with the exception of infilling tasks, for which we use temperature sampling with a temperature of 1.0. The maximum number of generated tokens is set to 32 for *Waiting Line* and 64 for both *Puzzle* and *Text Writing*.

**Evaluation Metrics**  For the *Waiting Line* and *Puzzle* tasks, we use accuracy, defined as a binary score indicating whether the generated output exactly matches one of the valid solutions. For the *Text Writing* benchmarks, we employ several scores consisting of a base grammar score and task-specific metrics. The grammar score, evaluated using the tool from Morris (2025), is 1 for grammatically correct outputs and 0 otherwise. The task-specific metrics are as follows: For *Summarization*, we report ROUGE-L (Lin, 2004) against a reference summary. For *Paraphrasing*, we measure both semantic similarity using BERTScore (Zhang et al., 2019) and lexical diversity using $1 - $ BLEU (Papineni et al., 2002) to penalize outputs that are too close to the original sentence. For *Words-to-Sentence*, we use an inclusion accuracy metric that verifies if all requested words are present in the generated sentence. For simplicity, we primarily report the grammar score in the main body of the paper. A comprehensive breakdown of all metrics is available in Appendix H.1.

**Dataset Generation**  The tasks for our benchmark were sourced and generated as follows. For *Waiting Line*, we created lists by randomly sampling from a predefined list of first and last names, which were generated with Gemini 2.5 Pro (Google Deepmind, 2025). For *Latin Square*, we generated prompts requesting the creation of Latin squares of size 4x4 with random symbols. For *Sudoku*, we used the existing dataset from Ye et al. (2025b). For *Summarization* and *Paraphrasing*, we utilized 100 examples of the existing SAMSum (Gliwa et al., 2019) and Vorobev & Kuznetsov (2023) benchmarks, respectively. Finally, for *Words-to-Sentence*, we used Gemini 2.5 Pro to generate word sets of four words at three difficulty levels: easy (simple, semantically aligned words), medium (more complex but aligned words), and hard (complex words that are challenging to connect meaningfully).

Table 3: Benchmark prompts and examples. **Bold** indicates the input part of the prompt.

| Task | Prompt | Example Answer |
|------|--------|----------------|
| Copy | You are managing a waiting line at a customer service desk. You need to record the following people in the order they arrived: **["Billy Ramos", "Alan Wells", "Grace Wright"]** Please copy the list exactly and provide only the final list. | ["Billy Ramos", "Alan Wells", "Grace Wright"] |
| Sort | You are managing a waiting line at a customer service desk. The following people should be organized alphabetically by last name for efficient processing: **["David Lewis", "Patrick Tran", "Sean Diaz"]** Please sort the list in alphabetical order and provide only the final list. | ["David Lewis", "Patrick Tran", "Sean Diaz"] |
| Reverse | You are managing a waiting line at a customer service desk. The previous staff member put the waiting line in the wrong order. Please reverse the order of the following people in the waiting line to correct it: **["Mark Richardson", "Kevin Martinez", "Henry Young"]** Please reverse the order of the list and provide only the final list. | ["Henry Young", "Kevin Martinez", "Mark Richardson"] |
| Shuffle | You are managing a waiting line at a customer service desk. The waiting line should be randomly shuffled to ensure fair service distribution: **["Paul Payne", "Robert Riley", "Peter Stone"]** Please randomly shuffle the list and provide only the final list. Ensure the sequence is different from the original. | ["Robert Riley", "Paul Payne", "Peter Stone"] |

| Task | Prompt | Example Answer |
|---|---|---|
| Replace Index | You are managing a waiting line at a customer service desk. The person at position **0** must be replaced with "**Henry Warren**": [**"Patrick Morgan", "Eric King", "Joe Reed"**] Please replace the person at the specified position with "**Henry Warren**" and provide only the final list. | ["Henry Warren", "Eric King", "Joe Reed"] |
| Replace Random | You are managing a waiting line at a customer service desk. One person in the waiting line must be replaced with "**Juan Torres**": [**"David Owens", "Kelly Payne", "Aaron Freeman"**] Please replace one random person with "**Juan Torres**" and provide only the final list. | ["David Owens", "Kelly Payne", "Juan Torres"] |
| Insert Index | You are managing a waiting line at a customer service desk. A new person "**Aaron Stewart**" is inserted into the line at position **2**: [**"Charlotte Chavez", "Grace Baker", "Keith Cooper"**] Please put the new person at the specified position and provide only the final list. | ["Charlotte Chavez", "Grace Baker", "Aaron Stewart", "Keith Cooper"] |
| Insert Random | You are managing a waiting line at a customer service desk. A new person "**Justin McDonald**" is inserted into the line at a random position: [**"Johnny Sullivan", "Ryan Baker", "Juan Wilson"**] Please put the new person in the random position and provide only the final list. | ["Justin McDonald", "Johnny Sullivan", "Ryan Baker", "Juan Wilson"] |
| Remove Index | You are managing a waiting line at a customer service desk. The person at position **2** has left the waiting line: [**"Sarah Robertson", "Maria Mitchell", "Donald Hughes"**] Please remove the person at the specified position and provide only the final list. | ["Sarah Robertson", "Maria Mitchell"] |
| Remove Random | You are managing a waiting line at a customer service desk. One person has left the waiting line: [**"Karen Kim", "Jerry Hall", "Jose Marshall"**] Please remove a random person and provide only the final list. | ["Jerry Hall", "Jose Marshall"] |
| Summarization | Summarize the following conversation. Only output the final result. **Steve: Bought the new Dream Theater album 5 minutes ago. I hope it's good. Rob: I have it here on my desk, ready for the first listening. Steve: Ok, I'll tell you later what I think about it. Rob: Same here. See you later!** Summary: | Steve and Rob will talk about new Dream Theater album, after they finish listening. |
| Paraphrasing | Paraphrase the following sentence. Only output the final result. Sentence: **Any idea of what sweater this is?** Paraphrase: | Can you identify this sweater? |
| Words-to-Sentence (easy) | Construct a single, coherent sentence using the words **dog, park, ball, and throw**. | I love to throw the ball for my dog at the park. |
| Words-to-Sentence (medium) | Construct a single, coherent sentence using the words **apple, river, table, and happy**. | I was so happy to sit at the table by the river and eat a crisp apple. |
| Words-to-Sentence (hard) | Construct a single, coherent sentence using the words **algorithm, river, symphony, and moss**. | The growth of the moss along the river bank seemed to follow a natural algorithm, a quiet symphony of life unfolding. |

| Task | Prompt | Example Answer |
|---|---|---|
| Latin Square (4x4) | Generate a Latin square of size **4** with the symbols **[H, 4, C, A]**. Only output the final result as CSV. | H,4,A,C 4,C,H,A C,A,4,H A,H,C,4 |
| Sudoku (4x4) | Fill the positions where the values are 0 in a 4x4 grid with digits 1-4 so that each column, each row, and each of the four 2x2 subgrids that compose the grid contains all of the digits from 1 to 4. Input: **0042 2031 4023 3214** Output: | 1342 2431 4123 3214 |

# D    ADDITIONAL EXPERIMENTAL RESULTS

## D.1    UNMASKING STRATEGIES: MARGIN TOP-K AND ENTROPY TOP-K

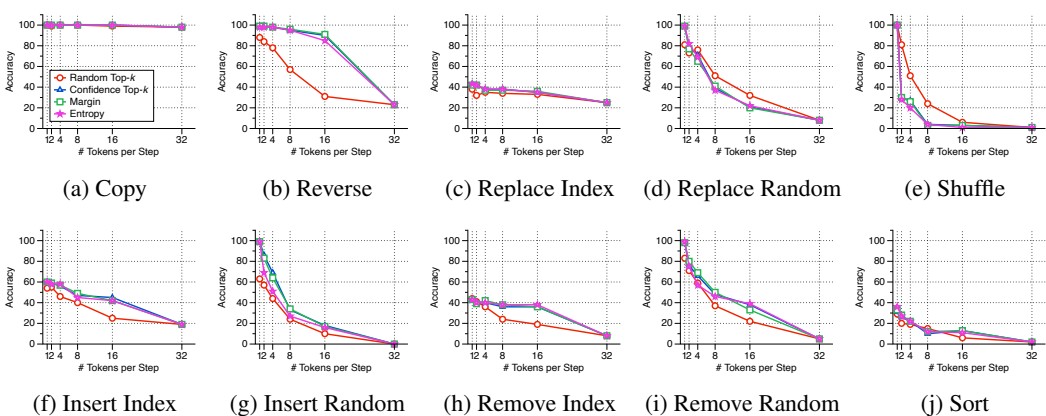

Figure 9: Full *Waiting Line* ($n = 3, 4, 5, 6$) results using LLaDA 1.5 (Zhu et al., 2025) with *Margin Top-k* and *Entropy Top-k*.

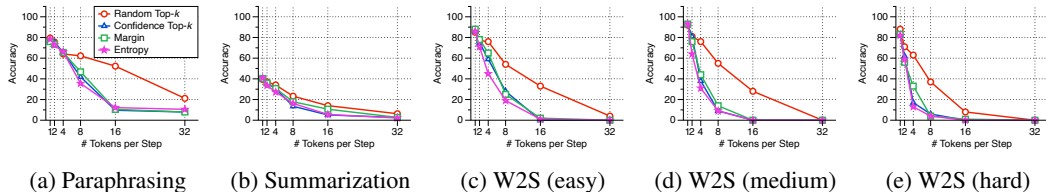

Figure 10: Full *Text Writing* results using LLaDA 1.5 (Zhu et al., 2025) with *Margin Top-k* and *Entropy Top-k*.

Recently, two alternatives to confidence-based unmasking have been proposed: *Margin Top-k* (Kim et al., 2025b) and *Entropy Top-k* (Ye et al., 2025b). These methods function similarly by iteratively revealing tokens but employ different metrics to quantify model uncertainty.

*Margin Top-k* calculates the score as the difference between the probabilities of the two most likely token candidates, $p_1$ and $p_2$. The margin, $m = p_1 - p_2$, is large when the model is highly confident in a single token, and small when there is ambiguity between the top candidates.

*Entropy Top-k*, in contrast, uses the Shannon entropy of the entire output probability distribution to measure uncertainty. A low entropy value signifies a peaked, high-confidence distribution, whereas a high entropy value indicates a more uniform and uncertain distribution.

We compare both approaches against *Confidence Top-k* in Fig. 9 and Fig. 10 to evaluate their relative performance. In our experiments, *Margin Top-k* and *Entropy Top-k* yield perform similarly to *Confidence Top-k*.

## D.2 FACTOR-BASED UNMASKING

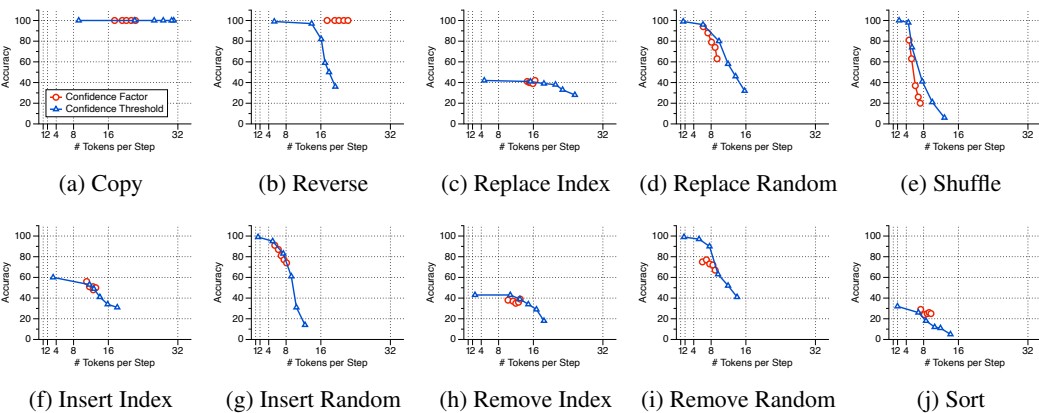

(a) Copy     (b) Reverse     (c) Replace Index     (d) Replace Random     (e) Shuffle

(f) Insert Index     (g) Insert Random     (h) Remove Index     (i) Remove Random     (j) Sort

Figure 11: Full *Waiting Line* ($n = 3, 4, 5, 6$) results using LLaDA 1.5 (Zhu et al., 2025) with *Factor-based* unmasking.

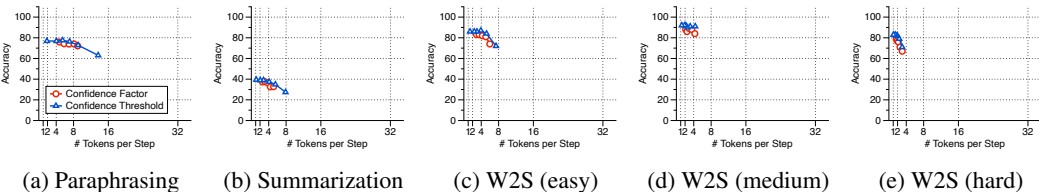

(a) Paraphrasing     (b) Summarization     (c) W2S (easy)     (d) W2S (medium)     (e) W2S (hard)

Figure 12: Full *Text Writing* results using LLaDA 1.5 (Zhu et al., 2025) with *Factor-based* unmasking.

*Factor-based* unmasking (Wu et al., 2025) is an adaptive method for determining how many tokens to decode in parallel during inference. After calculating the confidence scores for potential next tokens, the scores are sorted in descending order. The strategy then selects the largest number of tokens, $n$, that satisfies the condition:

$$(n + 1)(1 - c(n)) < f$$

Here, $c(n)$ represents the confidence of the $n$-th token in the sorted list, and $f$ is a predefined hyperparameter.

In contrast to *Threshold-based* unmasking, which accepts any token whose individual confidence exceeds a fixed value $\tau$, *Factor-based* is dynamic. It evaluates tokens as a group, allowing the degree of parallelism to increase when the model is highly confident and decrease when it is not.

In Fig. 11 and Fig. 12 we compare *Factor-based* unmasking with *Threshold-based* unmasking with the following values for $f \in \{0.7, 1.0, 1.3, 1.6, 1.9\}$. The results show that *Factor-based* unmasking follows a similar trend compared to *Threshold-based* unmasking. However, it has a narrower range of the number of tokens per step in general.

## D.3 IMPACT OF KV CACHING

Unlike autoregressive LLMs, dLLMs cannot effectively utilize KV cache, causing significant latency at each timestep. Recent training-free KV cache methods (Wu et al., 2025) claim dramatic speedups with minimal accuracy loss on benchmarks like GSM8K (Cobbe et al., 2021) and HumanEval (Chen et al., 2021). In Fig. 13 we evaluate whether PrefixCache from Fast-dLLM (Wu et al., 2025), maintains effectiveness on PARALLELBENCH under parallel decoding. The results show that caching decreased accuracy in general. Nevertheless it maintains a similar trend to no caching where parallel decoding decreases performance. Complete results for PrefixCache are in Fig. 32.

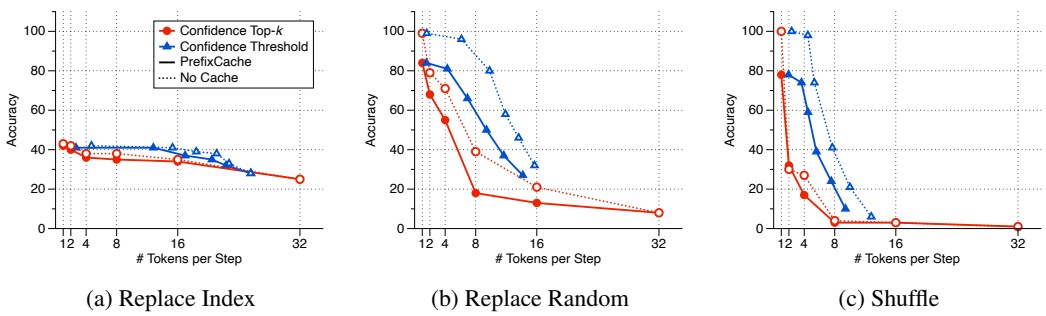

Figure 13: Fast-dLLM results on *Replace Index, Replace Random*, and *Shuffle* using PrefixCache.

### D.4 SEMI-AUTOREGRESSIVE DECODING

Semi-autoregressive decoding partitions the input sequence into blocks of equal length. These blocks are processed sequentially from left to right, while the tokens within each individual block are decoded in parallel. In the edge case where the block length is one, this process becomes equivalent to standard autoregressive decoding, generating a single token at a time. We investigate the impact of varying block lengths on performance for several unmasking methods, fixing the number of tokens decoded in parallel to $k = 2$ (Figs. 14 and 15).

In *Text Writing* tasks, performance improves as block length increases. This suggests that global dependencies are crucial for grammatical correctness, whereas smaller blocks often lead to errors and lower scores. Conversely, the *Waiting Line* tasks show an opposite trend. Here, left-to-right (decoding length 2) enhances performance by leveraging local dependencies to better match formatting tokens with the surrounding content.

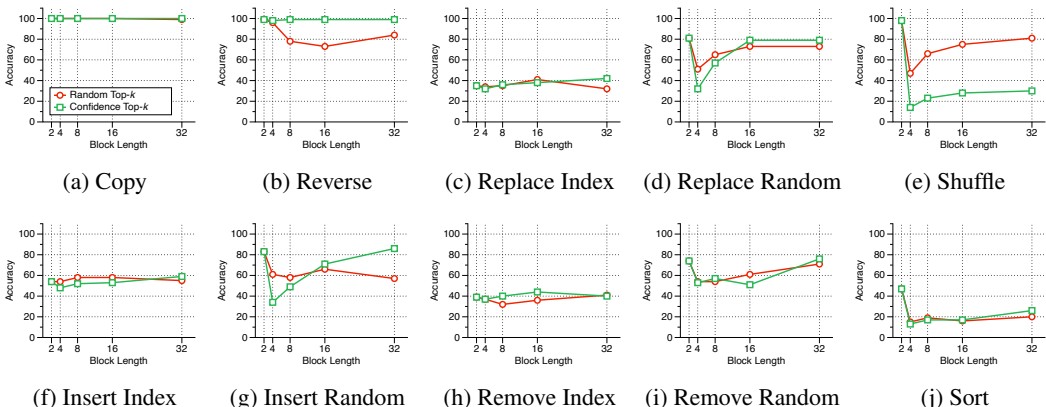

Figure 14: Full *Waiting Line* ($n = 3, 4, 5, 6$) results using LLaDA 1.5 (Zhu et al., 2025) with semi-autoregressive decoding ($k = 2$).

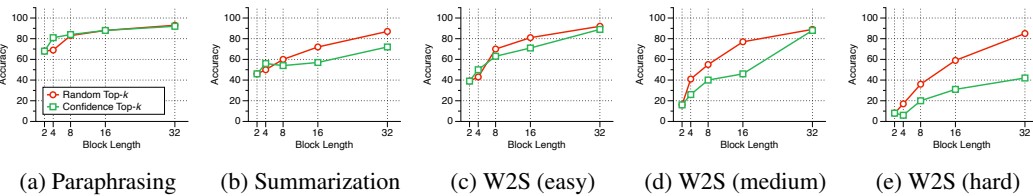

Figure 15: Full *Text Writing* results using LLaDA 1.5 (Zhu et al., 2025) with semi-autoregressive decoding ($k = 2$).

### D.5 BENCHMARK RESULTS ON LONGER OUTPUTS

We intentionally focus on short-output settings to isolate the effects of parallel decoding from model capacity limitations. In practice, even SOTA open-source dLLMs such as LLaDA and Dream achieve only around 50% one-by-one decoding accuracy on Waiting Line tasks when n increases, despite these tasks being conceptually simple. This indicates that longer outputs quickly introduce capacity bottlenecks that would confound our analysis. Importantly, PARALLELBENCH is designed so that output length can be freely adjusted through hyperparameters, enabling longer-output evaluations as future, higher-capacity dLLMs become available.

For completeness, we also report results on longer outputs. For *Waiting Line*, we increase $n$ from $\{3, 4, 5, 6\}$ to $\{21, 22, 23, 24\}$ and the max generation length from 32 to 128. For *Paraphrasing*, we use the XSum dataset and increase the generation limit from 64 to 256. For *Words-to-Sentence*, we expand from single-sentence generation ($n = 4$) to ten-sentence generation ($n = 7$), increasing the generation limit from 64 to 256.

As shown, except for very simple tasks such as *Copy*, one-by-one decoding already performs poorly in longer-output settings. Furthermore, even *Copy* undergoes sharp degradation as parallelism increases, while other tasks rapidly collapse to near-zero accuracy. For *Text Writing* tasks, both *Paraphrasing* and *Words-to-Sentence* similarly converge toward near-zero accuracy as parallelism increases. This suggests that limited model capacity significantly amplifies parallel decoding degradation for longer outputs.

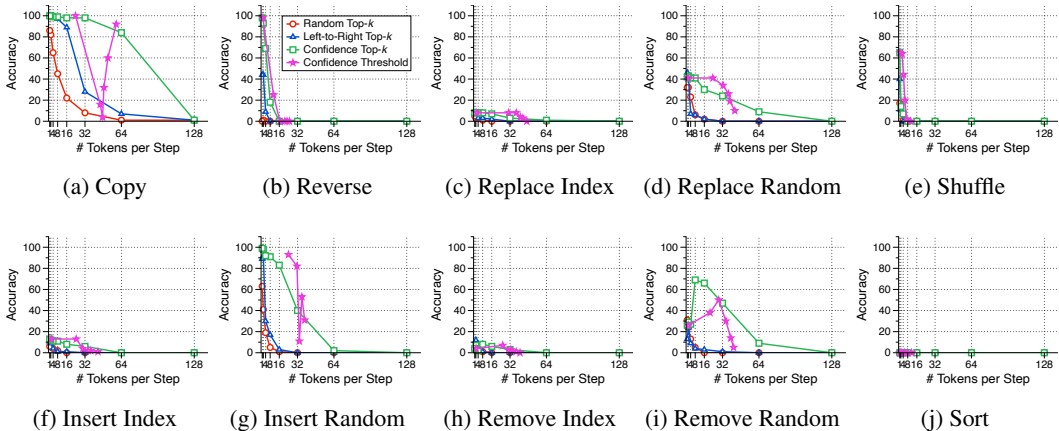

Figure 16: Longer *Waiting Line* ($n = 21, 22, 23, 24$) results using LLaDA 1.5 (Zhu et al., 2025).

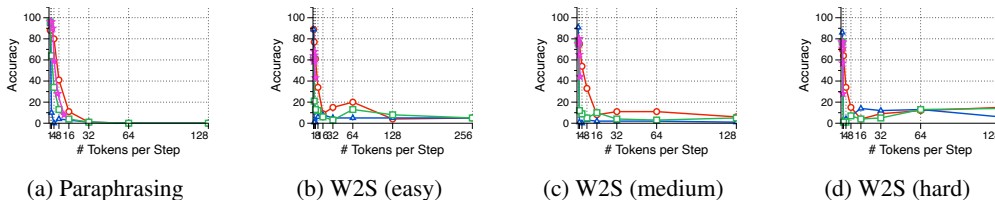

Figure 17: Longer *Text Writing* results using LLaDA 1.5 (Zhu et al., 2025).

### D.6 BENCHMARK RESULTS WITH ADDITIONAL METRICS

For *Text Writing* tasks, we also report the BERTScore (Zhang et al., 2019), ROUGE-L (Lin, 2004), and Inclusion Accuracy metrics introduced in Appendix C. BERTScore and ROUGE-L are substantially less sensitive to parallelism, showing only mild degradation as the number of tokens per step increases. By contrast, inclusion accuracy behaves similarly to the standard accuracy metric: as parallelism increases, models increasingly fail to include all required content, leading to a clear decline in inclusion accuracy.

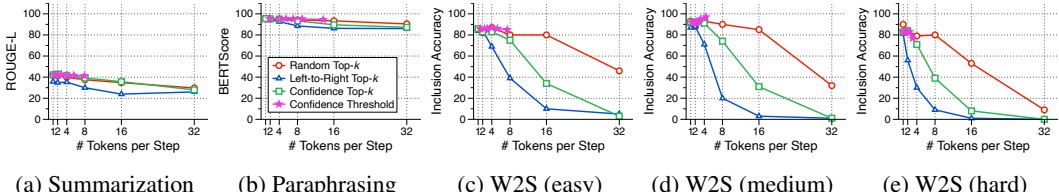

(a) Summarization  (b) Paraphrasing  (c) W2S (easy)  (d) W2S (medium)  (e) W2S (hard)

Figure 18: Benchmark results with additional metrics using LLaDA 1.5 (Zhu et al., 2025). We use ROUGE-L for *Summarization*, BERTScore for *Paraphrasing*, and an inclusion accuracy for *W2S*.

## E ADDITIONAL TECHNIQUES

### E.1 BENCHMARK RESULTS ON ADDITIONAL PARALLEL DECODING METHODS

We benchmark the speed-quality trade-offs of the newly released methods SlowFast Sampling (Wei et al., 2025), DUS (Luxembourg et al., 2025), WINO (Hong et al., 2025), and APD (Israel et al., 2025). Since APD currently supports only Dream, we report results for APD on Dream 7B and the rest on LLaDA 1.5.

SlowFast Sampling and DUS show worse trade-off curves than the confidence-threshold method on PARALLELBENCH, whereas WINO shows a better curve, indicating emerging progress toward intelligent parallel decoding. However, all remain far below the oracle, leaving substantial room for improvement. For APD, performance surpasses the confidence-threshold method at high parallelism but falls short of it at low parallelism, again suggesting considerable room for future advances.

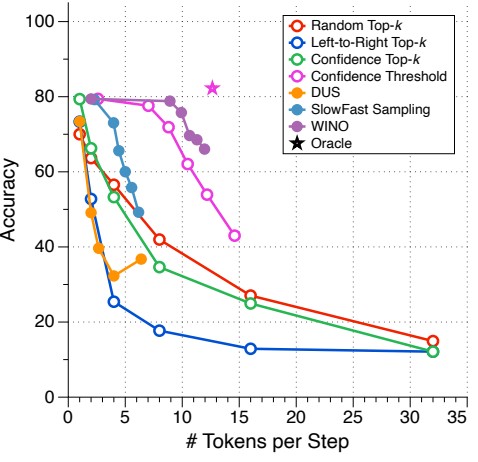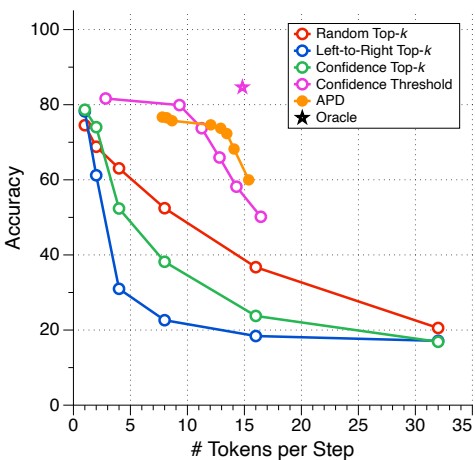

(a) Speed-quality trade-offs of additional parallel decoding methods evaluated on LLaDA 1.5.

(b) Speed-quality trade-offs of additional parallel decoding methods evaluated on Dream 7B.

Figure 19: Speed-quality trade-offs of additional parallel decoding methods.

### E.2 IMPACT OF FINE-TUNING

We fine-tuned the LLaDA 1.5 model on each task separately. For each task, we generated distinct training and validation sets of 20,000 and 5,000 examples, respectively. The model was trained for 10 epochs using the AdamW optimizer with a batch size of 32, a learning rate of $1 \times 10^{-5}$, a warmup rate of 0.05, and a cosine scheduler. We employed LoRA (Hu et al., 2022) for efficient training, configured with $r = 128$ and $\alpha = 256$, and applied it exclusively to the query, key, and value projection layers, with a dropout rate of 0.05. All experiments were conducted on a single A100 GPU and took approximately 2 hours to complete. As detailed in Figs. 20 and 21, fine-tuning improved overall performance but was insufficient to overcome tasks inherently challenging for parallel decoding, such as *Shuffle*, *Latin Square*, and *Insert/Remove/Replace Random*.

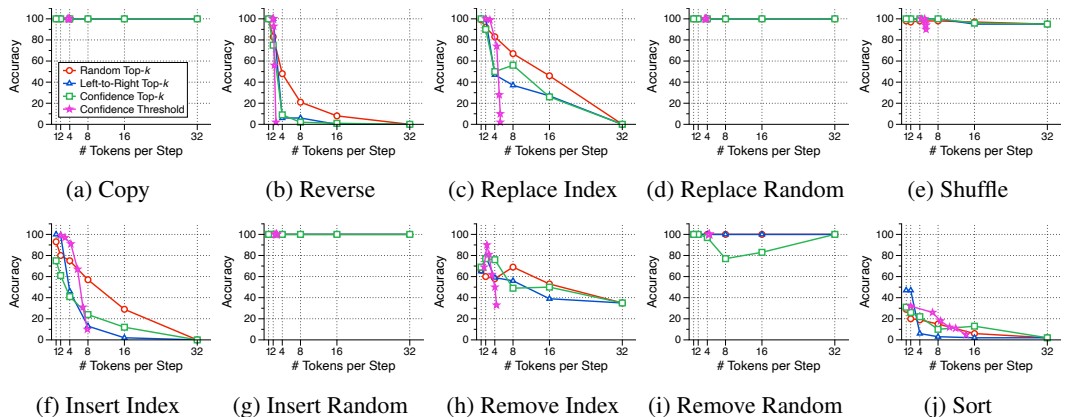

Figure 20: Full *Waiting Line* ($n = 3, 4, 5, 6$) results using fine-tuned LLaDA 1.5 (Zhu et al., 2025).

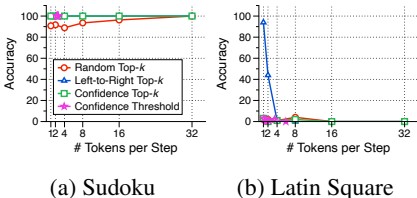

Figure 21: *Puzzle* results using fine-tuned LLaDA 1.5 (Zhu et al., 2025).

### E.3 CHAIN-OF-THOUGHT (COT) PROMPTING

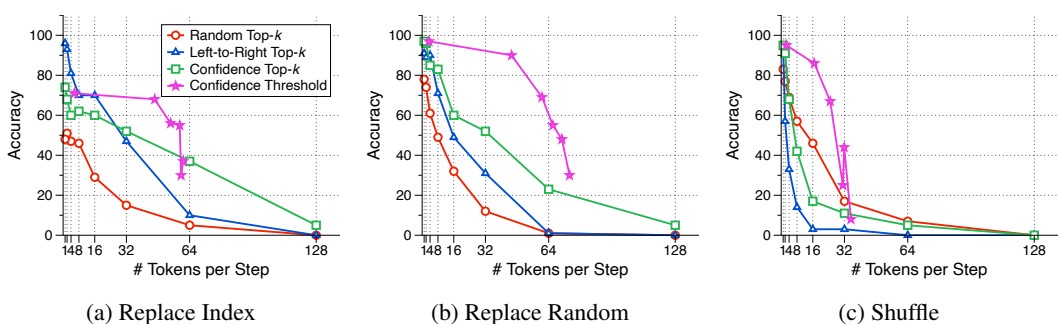

Figure 22: One-shot CoT results on *Replace Index, Replace Random*, and *Shuffle*. The one-shot examples for each sample are generated by GPT-4.1 Mini (OpenAI, 2025).

To investigate the impact of explicit reasoning on LLaDA 1.5's performance on the *Waiting Line* tasks, we tested Chain-of-Thought (CoT) prompting. The one-shot examples were generated using OpenAI's GPT-4.1-mini-2025-04-14 API (OpenAI, 2025).

The output token limit was increased from 32 to 256 to accommodate the verbose, multi-step nature of the CoT outputs. Our prompt was structured to include the trigger phrase, "Think step by step and give your final answer at the end as 'The answer is: {answer}'." after the user's question. The full prompt template is shown in Table 4.

As shown in Fig. 22, Chain-of-Thought (CoT) improves performance, especially for parallel decoding, but at the cost of a higher token count.

Table 4: An example of an one-shot example for *Shuffle* (CoT) generated with GPT-4.1 Mini (OpenAI, 2025).

| Question | You are managing a waiting line at a customer service desk. The waiting line should be randomly shuffled to ensure fair service distribution: ["Thomas Holmes", "Keith Ramos", "Victoria Collins", "Roger Hughes", "Austin Dunn"]. Please randomly shuffle the list. Ensure the sequence is different from the original. Think step by step and give your final answer at the end as 'The answer is: {answer}'. |
|---|---|
| Response | Step 1: Identify the original list: ["Thomas Holmes", "Keith Ramos", "Victoria Collins", "Roger Hughes", "Austin Dunn"]

Step 2: Randomly shuffle the list ensuring the new sequence is different from the original. One possible shuffled sequence could be: ["Austin Dunn", "Victoria Collins", "Roger Hughes", "Thomas Holmes", "Keith Ramos"]

Step 3: Verify the shuffled list is different from the original: Original first element: "Thomas Holmes" Shuffled first element: "Austin Dunn" They are different, so the shuffle is valid.

The answer is: ["Austin Dunn", "Victoria Collins", "Roger Hughes", "Thomas Holmes", "Keith Ramos"] |

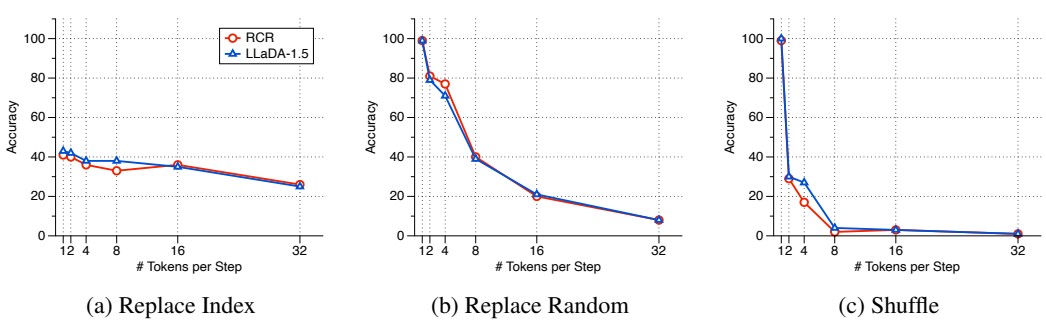

(a) Replace Index      (b) Replace Random      (c) Shuffle

Figure 23: RCR (He et al., 2025) results on *Replace Index, Replace Random*, and *Shuffle*.

### E.4 REMASKING SAMPLERS: RCR AND REMDM

We evaluated two training-free decoding strategies that allow for token revision: Running Confidence Remasking (RCR) (He et al., 2025) and Remasking Diffusion Model (ReMDM) (Wang et al., 2025), using their official codebases. RCR works by tracking the running maximum confidence for each position over time and continuously remasking tokens that remain uncertain. In contrast, ReMDM performs $n$ revision steps when most of the output is generated, targeting the $k$ tokens with the lowest confidence. For our experiments with ReMDM, we set $k = 1$ for simplicity and tested

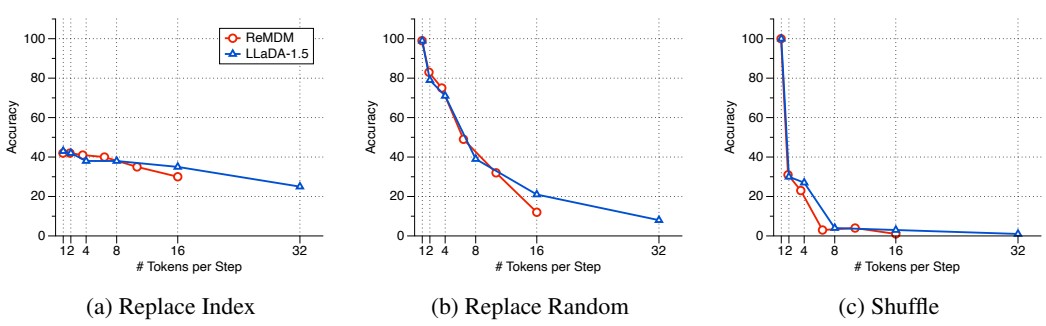

(a) Replace Index      (b) Replace Random      (c) Shuffle

Figure 24: ReMDM (Wang et al., 2025) results on *Replace Index, Replace Random*, and *Shuffle*.

various values for $n$. We found that a single revision step ($n = 1$) achieved optimal performance, with no further improvements observed from additional steps.

In our experiments (Figs. 23 and 24), we notice no significant improvements from RCR or ReMDM.

### E.5 Discrete Diffusion with Uniform Transition Matrix

This section evaluates two variants of the 90-million-parameter SEDD model (Lou et al., 2024): absorb and uniform. For the absorb variant, we employed the publicly available pre-trained model[3]. For the uniform variant, we pre-trained the model from scratch on the OpenWebText dataset (Gokaslan & Cohen, 2019). The pre-training process required 24 hours on eight A100 GPUs.

Subsequently, both variants were fine-tuned on each task. We trained for 32 epochs using the AdamW optimizer with a batch size of 64, a learning rate of $3 \times 10^{-4}$, and a cosine scheduler with a 0.25 warmup ratio. Each fine-tuning run was conducted on a single A100 GPU and completed in approximately one hour.

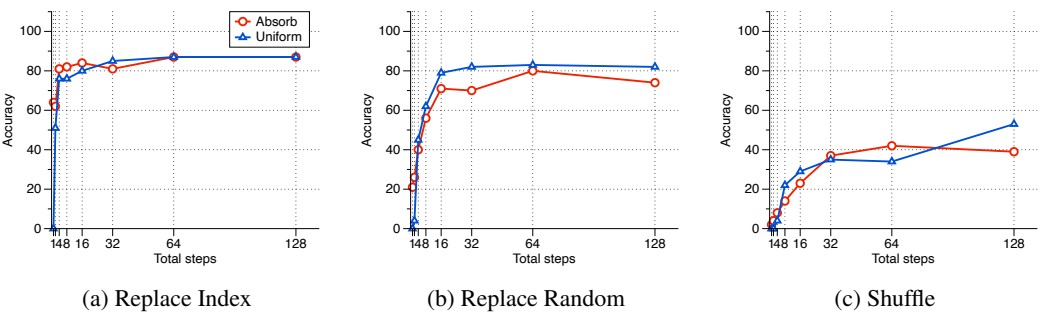

(a) Replace Index  (b) Replace Random  (c) Shuffle

Figure 25: SEDD (Lou et al., 2024) results on *Replace Index, Replace Random*, and *Shuffle*.

## F  Task Characterization

### F.1  Characterizing Tasks by Parallelizability and Decoding Order

We analyze how the unmasking order and the degree of parallel token decoding influence downstream scores for each task in our benchmark (Fig. 6). We plot the tasks in a 2D scatter plot where the x-axis represents parallelizability and the y-axis represents the benefit from different decoding orders. Tasks on the right are harder to parallelize than those on the left. Tasks on the top benefit from a semi-autoregressive order, while those on the bottom perform better with any-order decoding. The results are averaged over the LLaDA 1.5 and Dream 7B models and across four unmasking strategies: *Random Top-k*, *Confidence Top-k*, *Margin Top-k*, and *Entropy Top-k*.

To define the x-axis (parallelizability), we evaluate the model with an increasing number of parallel tokens, $k$, while using a full any-order decoding scheme. We then compute a metric representing the point at which performance begins to drop. This metric is the center of mass of the scores, calculated as $\frac{\sum_k (k \cdot \text{score}_k)}{\sum_k \text{score}_k}$. A smaller value on the x-axis therefore indicates that the task is harder to parallelize. We observe that the *Shuffle* task is the most difficult to parallelize. *Words-to-Sentence* is also challenging, as the model must successfully connect words and insert appropriate connecting language while maintaining grammatical correctness. Interestingly, the *Latin Square* task is harder to parallelize than *Sudoku*, potentially because multiple valid solutions exist, forcing the model to commit to one solution during generation. Similarly, *Insert Random* proves harder to parallelize than *Insert Index*.

To define the y-axis (influence of decoding order), we analyze how semi-autoregressive decoding affects performance. We fix the number of parallel tokens to $k = 2$ and evaluate the downstream metric for various block lengths. We then fit a line to these scores and use its slope as our metric. A positive slope indicates that performance increases as the block length decreases (enforcing a

---

[3]https://github.com/louaaron/Score-Entropy-Discrete-Diffusion

more autoregressive process), while a negative slope indicates that larger blocks or fully any-order decoding is preferable. Our results show that most tasks have a negative value, suggesting that the models generally prefer any-order decoding. Text generation tasks, in particular, fall into this category. We also include four existing benchmarks, GSM8K (Cobbe et al., 2021), HumanEval (Chen et al., 2021), IFEval (Zhou et al., 2023), and MATH (Hendrycks et al., 2021) in our analysis. Both tasks appear on the left side of the plot, indicating that models can solve them in a more parallel fashion. This finding underlines the need for more challenging benchmarks like the proposed PARALLELBENCH to properly test the capabilities of dLLMs.

Finally, Tables 5 and 6 provide a comprehensive taxonomy of the evaluated tasks, characterizing them by their ideal token dependency and the scope of their constraints. As detailed in Table 5, *Waiting Line* tasks primarily exhibit zero dependency for deterministic tasks, though tasks involving randomness or shuffling introduce global constraints that significantly increase token dependency. In contrast, Table 6 highlights how *Text Writing* tasks maintain local constraints but scale in dependency from low to very high as the difficulty increases. The *Puzzles* category reflects a similar divergence: *Sudoku* is treated as a deterministic task with zero dependency, while the *Latin Square* task allows for multiple valid solutions, thereby introducing high dependency.

Table 5: *Waiting Line*: Dependencies and Constraints.

| Category | Waiting Line | | | | | | | | | |
|---|---|---|---|---|---|---|---|---|---|---|
| Task | Copy | Reverse | Replace Index | Replace Random | Shuffle | Insert Index | Insert Random | Remove Index | Remove Random | Sort |
| (Ideal) Token Dependency | Zero | Zero | Zero | Medium | Very High | Zero | Medium | Zero | Medium | Zero |
| Constraint Type | N/A | N/A | N/A | Global | Global | N/A | Global | N/A | Global | N/A |

Table 6: *Text Writing* and *Puzzles*: Dependencies and Constraints.

| Category | Text Writing | | | | | Puzzles | |
|---|---|---|---|---|---|---|---|
| Task | Para-phrasing | Summari-zation | W2S (Easy) | W2S (Medium) | W2S (Hard) | Sudoku | Latin Square |
| (Ideal) Token Dependency | Low | Low | Medium | High | Very High | Zero | Very High |
| Constraint Type | Local | Local | Local | Local | Local | N/A | Global |

## F.2 SPEED-QUALITY TRADE-OFF

Figure 7 illustrates the trade-off between the number of tokens decoded in parallel and the resulting downstream accuracy for each unmasking method. For our experiments, we utilized the LLaDA 1.5 model (Zhu et al., 2025). The accuracy is averaged over all 17 benchmark tasks across the three categories: *Waiting Line*, *Text Writing*, and *Puzzles*.

The oracle performance, marked by a star in the plot, represents an empirical upper bound. To compute this, we employ the *Confidence Threshold* method. For each sample, we identify the minimum threshold (equivalent to the minimum number of total performed steps) that achieves the best possible accuracy. We then average the accuracy and number of steps of these optimal thresholds across all samples and tasks. This simulates a hypothetical unmasking strategy that knows in advance the precise minimum effort required to solve each task perfectly.

The results indicate that for *Top-k* unmasking methods, accuracy degrades rapidly as the number of tokens per step increases. *Confidence Threshold*, on the other hand, exhibits a much more favorable curve, where the drop-off in accuracy occurs at a significantly higher number of parallel tokens. Nevertheless, a substantial gap remains when compared to the oracle. While all methods perform comparably when unmasking a single token per step, the oracle maintains high accuracy at a far greater decoding parallelism. This suggests that, in theory, unmasking could be performed much faster without sacrificing accuracy if a more effective sampling method were available.

### F.3 SPEED-QUALITY TRADE-OFF IN EXISTING BENCHMARKS

In this section, we assess the behaviour of parallel decoding on tasks not specifically designed for benchmarking parallel decoding performance, reporting speed–quality trade-offs. Fig. 26 shows the results for GSM8K (Cobbe et al., 2021), IFEval (Zhou et al., 2023), and MATH (Hendrycks et al., 2021), using LLaDA 1.5 with varying degrees of parallelism. All benchmarks exhibit a similar trend of decreasing accuracy as the number of tokens per step increases. Even when decoding one token per step, these benchmarks already exhibit substantial difficulty, which implies that model capacity remains a dominant factor that cannot be separated from the analysis. In contrast to our benchmark tasks, all existing tasks show slight accuracy degradation as parallelism increases. Due to these two properties, existing benchmarks do not permit a clear evaluation of whether a parallel decoding method can effectively control or adapt its level of parallelism in relation to task difficulty.

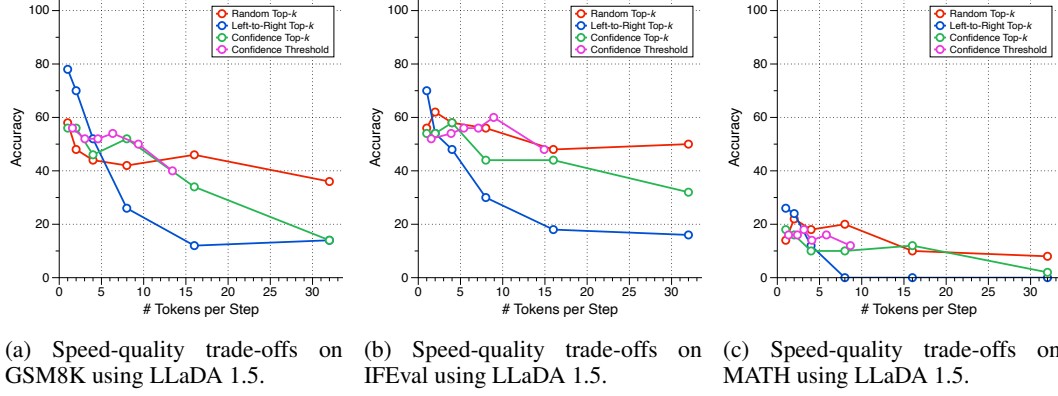

(a) Speed-quality trade-offs on GSM8K using LLaDA 1.5.

(b) Speed-quality trade-offs on IFEval using LLaDA 1.5.

(c) Speed-quality trade-offs on MATH using LLaDA 1.5.

Figure 26: Speed-quality trade-offs of existing benchmarks.

### F.4 TIME-QUALITY TRADE-OFF

We provide actual wall-clock latency measurements and their corresponding time–quality curves in Fig. 27. We obtained these results using a single A100 80GB GPU with a batch size of 1.

Notably, the time-quality curve mirrors the speed–quality curve shown in Fig. 7 (where the x-axis represents tokens per step). This indicates that analyzing the trade-off using tokens per step rather than hardware-dependent latency measurements is sufficient and reliable.

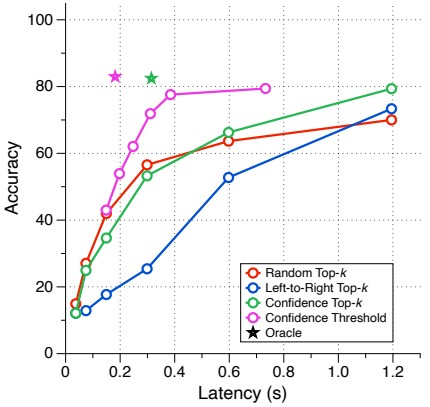

Figure 27: Time-quality trade-off.

## G    FAILURE CASES

We present several failure cases observed when using LLaDA 1.5 (Zhu et al., 2025) with *Confidence Top-k* ($k = 2$). These examples highlight specific weaknesses inherent to parallel decoding.

**Incorrect Swapping of Names in *Waiting Line (Shuffle)***

**Input:** ["Jack Burns", "Jessica Martin", "Tyler Lopez"]

**Model Output:** ["Jessica Lopez", "Jack Burns", "Tyler Martin"]

**Analysis:** The model incorrectly swaps the first and last names of two persons. This flaw highlights the risk of the conditional independence assumption in dLLMs.

**Word Repetition in *Summarization***

**Input:** "Summarize the following conversation. [...] Daniel: Cool."

**Model Output:** "Missy gets out of work at 6 pm and she drinks drinks after dinner."

**Analysis:** The model incorrectly repeats the word "drinks." This is a common artifact in parallel decoding, where the lack of one-by-one conditioning can cause the model to generate the same token in adjacent positions.

**Grammatical Error in *Paraphrasing***

**Input:** "Paraphrase the sentence: 'How do I deal with a rude student?'"

**Model Output:** "What strategies can I use to handle an disrespectful student?"

**Analysis:** The output contains a grammatical error in "an disrespectful". This demonstrates that the model fails to enforce local syntactic rules, as the choice of article depends on the subsequent token, a dependency that is weakened in a parallel generation scheme.

**Grammatical Error in *Words-to-Sentence***

**Input:** "Construct a single, coherent sentence using the words ball, bridge, elephant, and open."

**Model Output:** "The elephant walked across the the old bridge, looking carrying an open ball."

**Analysis:** The generated sentence is syntactically flawed, due to both a duplicated article ("the the") and a wrong verb sequence ("looking carrying"). This failure to form a coherent clause underscores the difficulty parallel decoding has in maintaining long-range grammatical structure.

## H    COMPREHENSIVE EXPERIMENTAL RESULTS

### H.1    COMPARISON WITH LARGE LANGUAGE MODELS

Tables 7 to 9 present a performance comparison between open-source dLLMs, a closed-source dLLM Mercury (Inception Labs et al., 2025), and several popular autoregressive LLMs, including Qwen (Yang et al., 2024; 2025a), Llama (Grattafiori et al., 2024), and Claude Haiku 3.5 (Anthropic, 2024). To create a more challenging benchmark for this comparison, we increased the list length of the *Waiting Line* task to $n = 15$ with 128 output tokens. For the open-source dLLMs, we set the number of tokens decoded in parallel to $k = 2$ and did not use semi-autoregressive decoding. In contrast, we had no control over Mercury's parallel decoding settings, which we assume are handled internally.

The results indicate that even a commercial model like Mercury cannot achieve a perfect score on *Shuffle*, yet it successfully solves *Reverse* with 100% accuracy. Furthermore, the relatively simple task of producing a *Latin Square* proved rather challenging for Mercury, whereas it was easier for most LLMs that decode tokens one by one. Nevertheless, Mercury exceeds or matches the performance of open-source models on most tasks.

Table 7: Benchmark results for *Waiting Line*.

| Model / Dataset | Waiting Line | | | | | | | | | |
|---|---|---|---|---|---|---|---|---|---|---|
| | Copy | Sort | Rev. | Shuff. | Ins. Ind. | Ins. Rand. | Rem. Ind. | Rem. Rand. | Rep. Ind. | Rep. Rand. |
| LLaDA (Nie et al., 2025) | **100.0** | **0.0** | 97.0 | 21.0 | 14.0 | 85.0 | 18.0 | 82.0 | 15.0 | 58.0 |
| LLaDA 1.5 (Zhu et al., 2025) | **100.0** | **0.0** | 99.0 | 18.0 | 14.0 | 86.0 | 21.0 | 77.0 | 15.0 | 70.0 |
| Dream 7B (Ye et al., 2025b) | **100.0** | **0.0** | 91.0 | 87.0 | 27.0 | 81.0 | **32.0** | 62.0 | 27.0 | **99.0** |
| DiffuCoder (Gong et al., 2025) | 58.0 | **0.0** | 28.0 | 32.0 | 9.0 | 51.0 | 10.0 | 35.0 | 19.0 | 57.0 |
| Mercury (Inception Labs et al., 2025) | **100.0** | **0.0** | 100.0 | 92.0 | 63.0 | 95.0 | 15.0 | 90.0 | 28.0 | 98.0 |
| Qwen2.5 3B (Yang et al., 2024) | **100.0** | 0.0 | 4.0 | 10.0 | 39.0 | 96.0 | 19.0 | 72.0 | 19.0 | 53.0 |
| Qwen2.5 7B (Yang et al., 2024) | **100.0** | 1.0 | 65.0 | 85.0 | 39.0 | **100.0** | 11.0 | 73.0 | 16.0 | 88.0 |
| Qwen3 4B (Yang et al., 2025a) | **100.0** | 0.0 | 75.0 | 97.0 | **77.0** | 89.0 | **43.0** | 47.0 | **67.0** | 92.0 |
| Llama 3.1 8B (Grattafiori et al., 2024) | **100.0** | **64.0** | 100.0 | 96.0 | 50.0 | 93.0 | 22.0 | 63.0 | 23.0 | 88.0 |
| Llama 3.2 3B (Grattafiori et al., 2024) | **100.0** | 30.0 | 83.0 | 92.0 | 20.0 | 35.0 | 17.0 | 64.0 | 13.0 | 70.0 |
| Claude Haiku 3.5 (Anthropic, 2024) | **100.0** | 5.0 | 100.0 | 100.0 | 31.0 | **100.0** | 29.0 | 100.0 | 32.0 | **100.0** |

Table 8: Benchmark results for *Text Writing*.

| Model / Dataset | Text Writing | | | | | | | | | | |
|---|---|---|---|---|---|---|---|---|---|---|---|
| | Paraphrasing | | | Summarization | | W2S (easy) | | W2S (medium) | | W2S (hard) | |
| | Grammar | BERT | 1-BLEU | Grammar | ROUGE-L | Grammar | Acc. | Grammar | Acc. | Grammar | Acc. |
| LLaDA (Nie et al., 2025) | 96.0 | 95.2 | 82.1 | 85.0 | **43.2** | 86.0 | 80.0 | 92.0 | 90.0 | 73.0 | 82.0 |
| LLaDA 1.5 (Zhu et al., 2025) | 95.0 | 95.2 | 83.4 | 77.0 | 42.9 | 89.0 | **83.0** | 88.0 | 92.0 | 72.0 | **84.0** |
| Dream 7B (Ye et al., 2025b) | 91.0 | 93.5 | 52.5 | 85.0 | 40.3 | 89.0 | 61.0 | 75.0 | 75.0 | 56.0 | 83.0 |
| DiffuCoder (Gong et al., 2025) | 86.0 | 94.2 | 54.7 | 78.0 | 41.9 | 87.0 | 53.0 | 87.0 | 72.0 | 73.0 | 59.0 |
| Mercury (Inception Labs et al., 2025) | **98.0** | 95.2 | **84.4** | 93.0 | 31.1 | 99.0 | 80.0 | 100.0 | 94.0 | 99.0 | **84.0** |
| Qwen2.5 3B (Yang et al., 2024) | **100.0** | 94.7 | 84.8 | 95.0 | 38.2 | 99.0 | 54.0 | 99.0 | 80.0 | 96.0 | 73.0 |
| Qwen2.5 7B (Yang et al., 2024) | 99.0 | 95.5 | 83.5 | 97.0 | 38.9 | 99.0 | 71.0 | 95.0 | 82.0 | 98.0 | 75.0 |
| Qwen3 4B (Yang et al., 2025a) | 98.0 | 94.9 | 86.2 | 93.0 | 34.8 | 97.0 | 78.0 | 95.0 | 81.0 | 99.0 | 86.0 |
| Llama 3.1 8B (Grattafiori et al., 2024) | 99.0 | 93.3 | 92.0 | 97.0 | **39.4** | 97.0 | 92.0 | 96.0 | **98.0** | 99.0 | **90.0** |
| Llama 3.2 3B (Grattafiori et al., 2024) | 99.0 | 92.9 | 88.8 | 97.0 | 36.4 | 98.0 | 75.0 | 96.0 | 94.0 | 99.0 | 70.0 |
| Claude Haiku 3.5 (Anthropic, 2024) | 99.0 | 92.3 | **96.2** | 99.0 | 34.3 | **100.0** | 93.0 | 98.0 | 97.0 | 99.0 | 84.0 |

Table 9: Benchmark results for *Puzzle*.

| Model / Dataset | Puzzle | |
|---|---|---|
| | Sudoku | Latin Square |
| LLaDA (Nie et al., 2025) | 14.8 | 36.0 |
| LLaDA 1.5 (Zhu et al., 2025) | 30.6 | 35.3 |
| Dream 7B (Ye et al., 2025b) | **95.4** | 34.7 |
| DiffuCoder (Gong et al., 2025) | 16.7 | 10.0 |
| Mercury (Inception Labs et al., 2025) | 11.1 | **59.0** |
| Qwen2.5 3B (Yang et al., 2024) | 2.8 | 16.0 |
| Qwen2.5 7B (Yang et al., 2024) | 1.9 | 88.0 |
| Qwen3 4B (Yang et al., 2025a) | 0.0 | 90.0 |
| Llama 3.1 8B (Grattafiori et al., 2024) | 0.0 | 24.0 |
| Llama 3.2 3B (Grattafiori et al., 2024) | 0.0 | 12.0 |
| Claude Haiku 3.5 (Anthropic, 2024) | 21.3 | **98.0** |

## H.2 COMPLETE BENCHMARK RESULTS

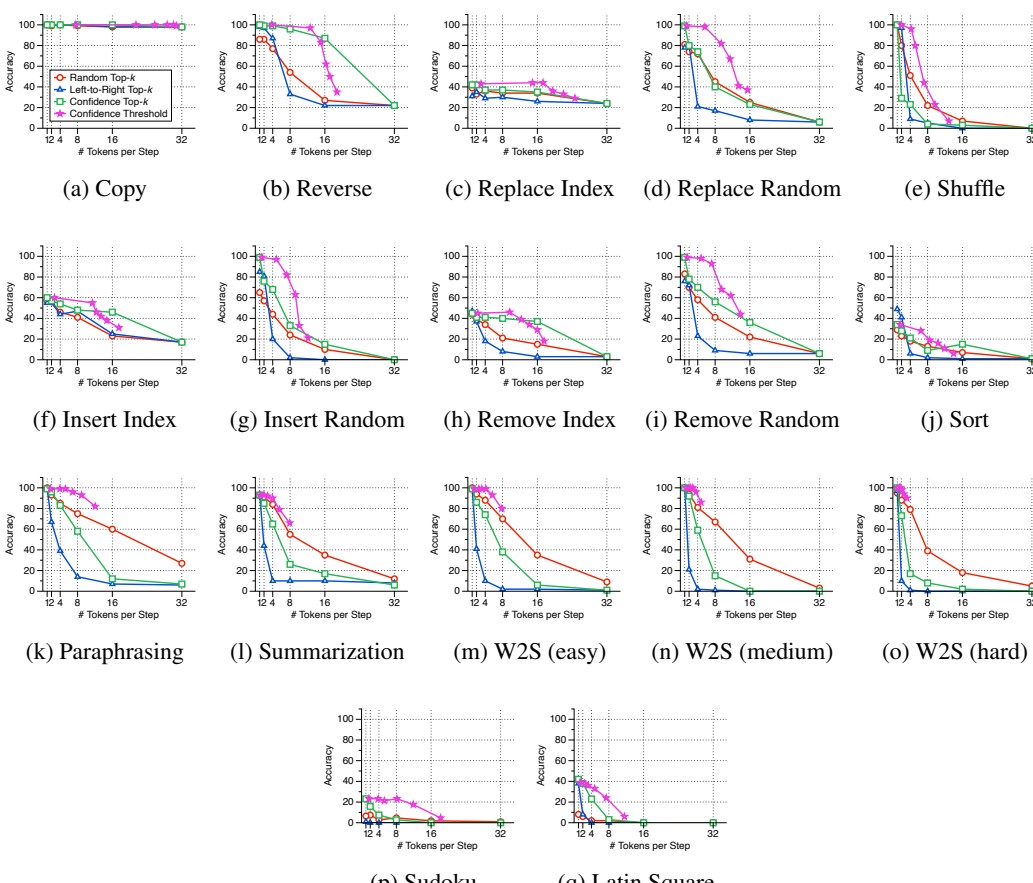

Figure 28: Full PARALLELBENCH results using LLaDA 1.0 (Nie et al., 2025).

## I AUTOREGRESSIVE LLM PARALLEL DECODING

In this section, we demonstrate that PARALLELBENCH generalizes beyond Diffusion LLMs to other parallel decoding paradigms. We examine Medusa (Cai et al., 2024), a speculative decoding framework that augments the frozen backbone of an autoregressive LLM with multiple lightweight decoding heads. Instead of relying on a separate draft model, these heads simultaneously predict multiple future tokens based on the current step's hidden states. Candidates are then organized into a tree structure and verified in a single parallel forward pass using Tree Attention, accepting the longest valid prefix.

To analyze the accuracy of parallel predictions across varying difficulty levels, we modify the Medusa inference process. Rather than utilizing the dynamic verification mechanism, we force the model to decode a fixed number of tokens per step (ranging from 1 to 5). We employ Vicuna-1.5-7B (Chiang et al., 2023) as the backbone.

Fig. 33 illustrates the performance across benchmark tasks. We observe a significant drop in quality when increasing the number of parallel tokens from 1 to 2, affecting even simple tasks like *Copy*. Furthermore, complex tasks such as *Puzzles* generally exhibit very low performance. These results confirm that PARALLELBENCH effectively evaluates arbitrary parallel decoding architectures, highlighting the challenges of parallelization even in autoregressive LLM settings.

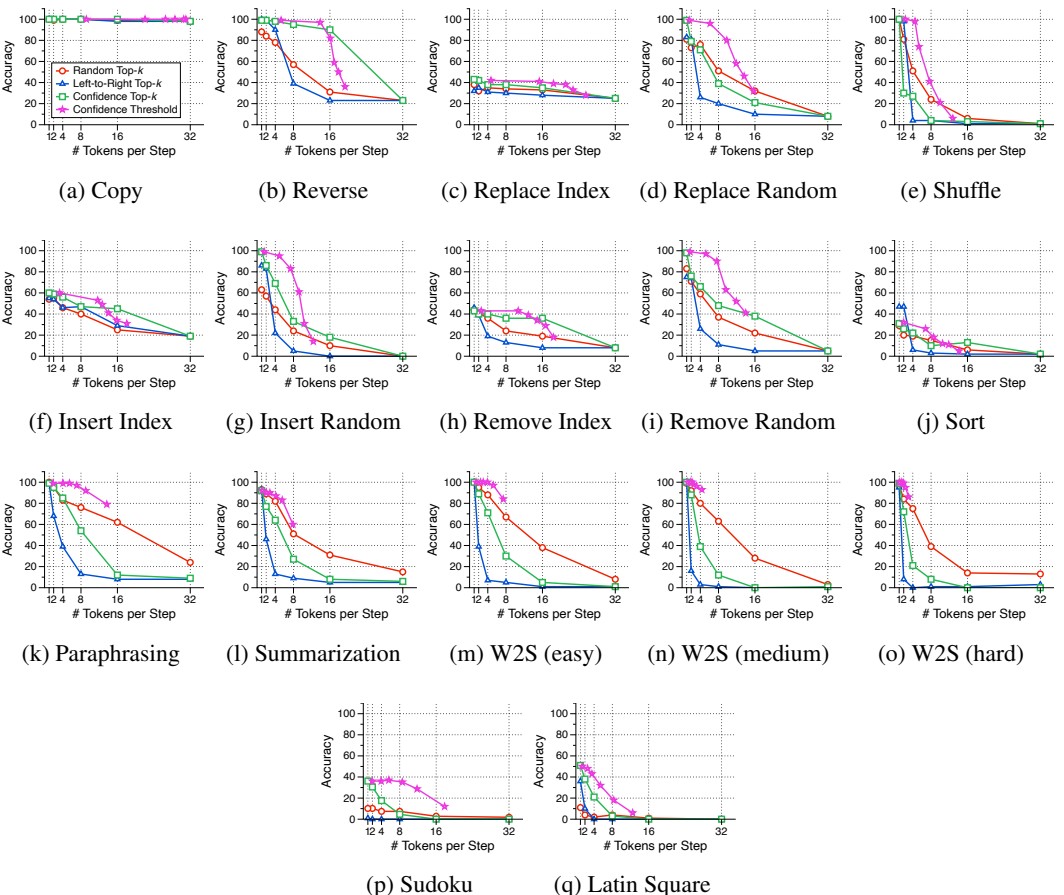

Figure 29: Full PARALLELBENCH results using LLaDA 1.5 (Zhu et al., 2025).

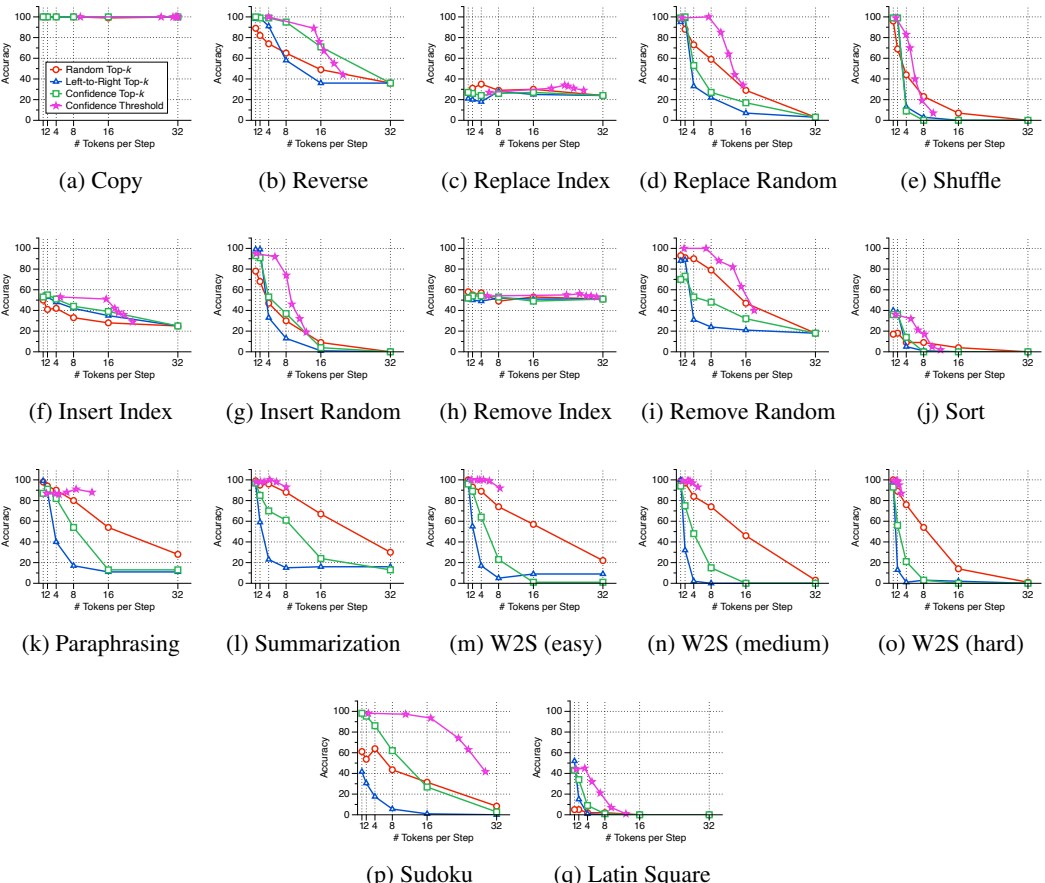

Figure 30: Full PARALLELBENCH results using Dream 7B (Ye et al., 2025b).

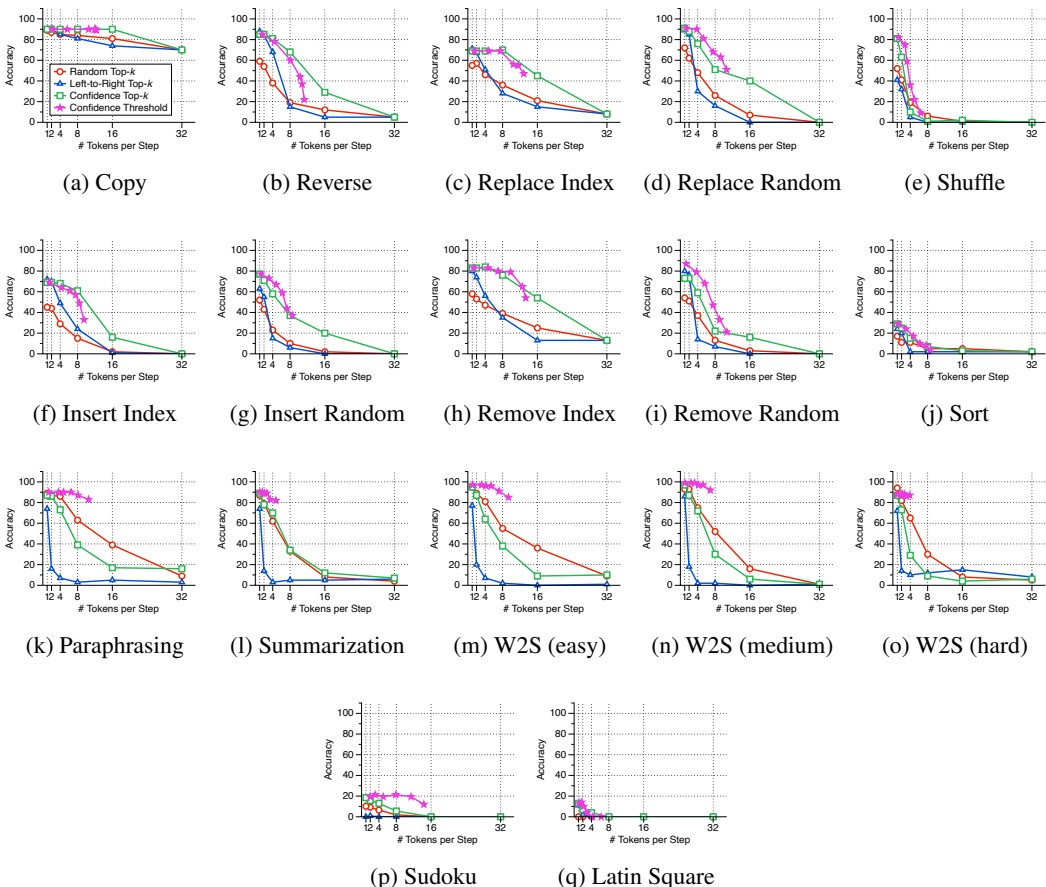

Figure 31: Full PARALLELBENCH results using DiffuCoder (Gong et al., 2025).

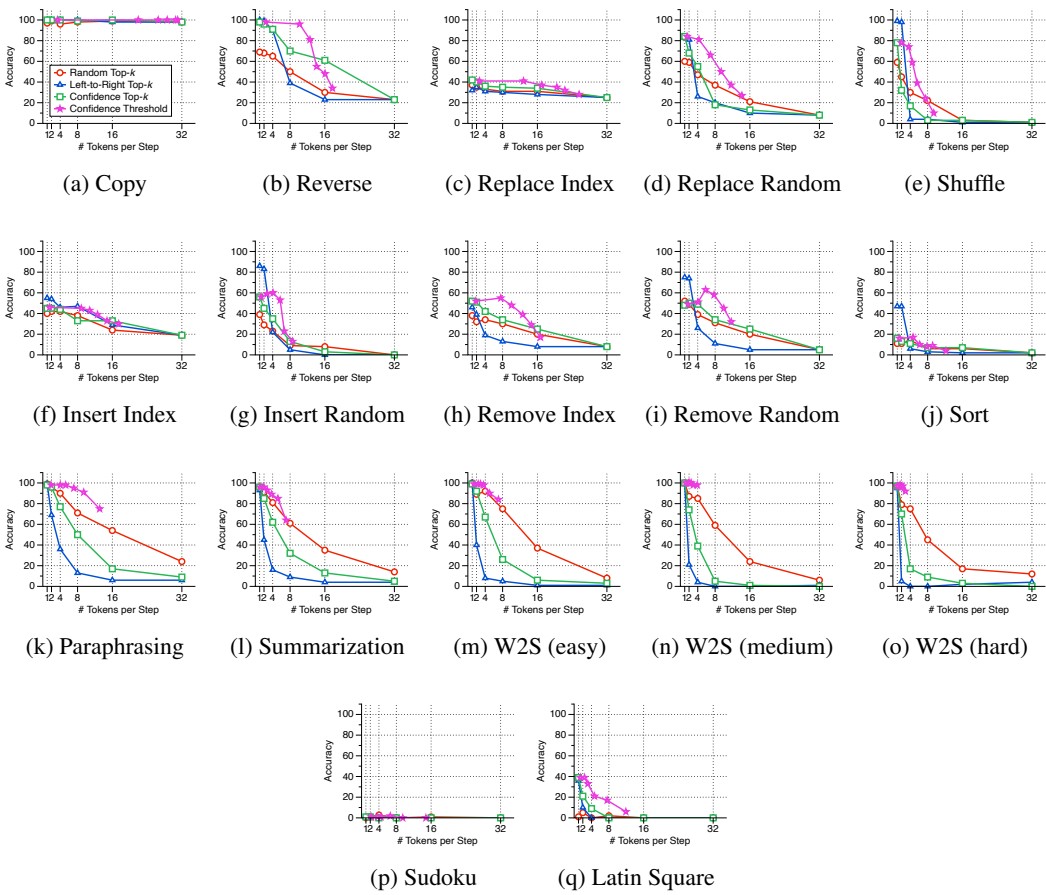

Figure 32: Full PARALLELBENCH results using LLaDA 1.5 (Zhu et al., 2025) with PrefixCache (Wu et al., 2025).

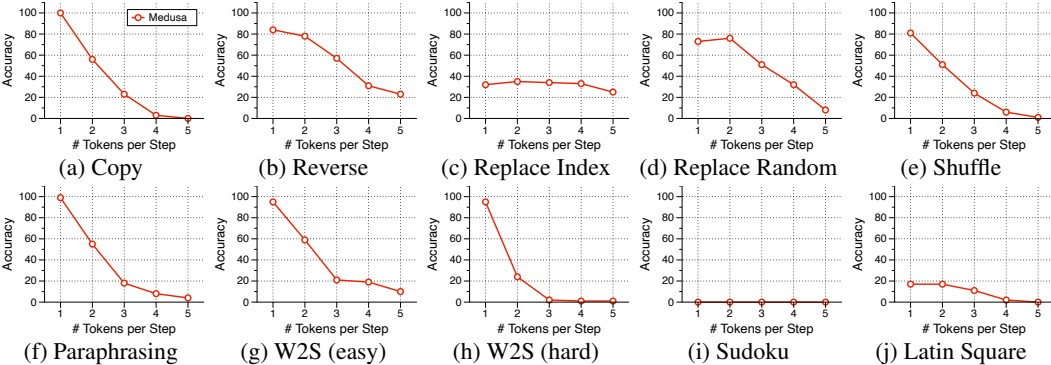

Figure 33: Benchmark results of Medusa (Cai et al., 2024) on PARALLELBENCH: *Waiting Line* (Figs. 33a to 33e), *Text Writing* (Figs. 33f to 33h), and *Puzzles* (Figs. 33i and 33j).

## LLM USAGE DISCLOSURE

We utilized Gemini 2.5 Pro (Google Deepmind, 2025) and GPT-4.1 Mini (OpenAI, 2025) to generate our benchmark dataset. Additionally, Gemini 2.5 Pro was used to assist with improving the grammar, clarity, and readability of this manuscript. The authors reviewed and edited all LLM-generated content and suggestions to ensure the final text accurately reflects our scientific contributions and claims. The authors retain full responsibility for the content of this paper.

