# OpenReview forum: "ParallelBench: Understanding the Trade-offs of Parallel Decoding in Diffusion LLMs"
_ICLR.cc/2026/Conference — ICLR 2026 Poster_

### Official Review · Reviewer_vHaA · 2025-10-30

**Soundness:** 2
**Presentation:** 2
**Contribution:** 2
**Rating:** 4
**Confidence:** 3

**Summary:**

The paper presented an information-theoretic analysis on the capabilities of diffusion LLMs using toy tasks to compute their theoretic bounds. The paper performed a distribution based analysis and a decoding strategy based analysis. The paper also presented ParallelBench, composed of the toy tasks and realistic tasks, to evaluate the capabilities of dllms empirically.

**Strengths:**

- The paper presents a theoretical explanation for the capabilities and limitations of parallel decoding with dLLMs. This provides some intuition for whether to use dLLMs or whether to enable decoding multiple tokens per step depending on the task at hand. The theoretical result is corroborated with empirical results on the same toy tasks.
- The empirical evaluation performed a wide variety of ablations that provide some insights on how different unmasking techniques and models compare.
- The presented benchmark can serve as a baseline for evaluating whether a particular language model/decoding technique can adaptively exploit parallelism, beyond dLLMs.

**Weaknesses:**

- The empirical evaluation on realistic tasks seem to deviate from the expectations set by the theoretical results. For example LLaDA 1.5 performs better on Latin Square than Sudoku, but Sudoku has C(Y|X) = 0 while Latin Square has C(Y|X) > 0. This weakens the significance of the theoretic analysis. The difference in complexity from the toy tasks to realistic tasks seem dominate the difference in task type.
- The empirical results for 3f-j show accuracy going down as the # tokens per step increase, which is to be expected that the more naive parallelism (as in the technique does not explicitly target/decide when parallelism is appropriate) employed the worse the accuracy would be. Most realistic tasks should fall into these regimes, rather than the extremes of Copy or Replace Index. And it is unlikely to be possible to perform the same information-theoretic analysis on realistic tasks to quantify the theoretic accuracy.
- The best the analysis provides is an intuition, which in general can be useful. However, the intuition that more dependencies between tokens in the expected response makes parallelism harder is hardly surprising. The difficult task is knowing when parallelism can be enabled, which the work does not provide directions towards any solutions, as the toy examples shown do not generalize/scale to real examples.

**Questions:**

- How is it that Sudoku having supposedly C(Y|X) = 0 and Latin Square having C(Y|X) > 0, but the empirical results showing that LLaDA 1.5 performs better on Latin square than Sudoku?
- On Figure 7, what is the arrow indicative of?
- Is there a reason why ParallelBench is for evaluating dLLM specifically and not parallel decoding in different LLMS (e.g. parallel decoding in AR LLMs) in general?

---

> ### Author Response · Authors · 2025-11-20
>
> We thank the reviewer for noting that (i) our paper presents a theoretical explanation that provides intuition, (ii) the theoretical results are corroborated by empirical evidence, (iii) our empirical evaluation offers useful insights, and (iv) our benchmark can serve as a baseline for evaluating parallel decoding strategies.
>
> ---
>
> > **W1, Q1: The empirical evaluation on realistic tasks seem to deviate from the expectations set by the theoretical results. How is it that Sudoku having supposedly $C(Y|X) = 0$ and Latin Square having $C(Y|X) > 0$, but the empirical results showing that LLaDA 1.5 performs better on Latin square than Sudoku?**
>
> Thank you for raising this important question.
>
> Our theory shows that under the conditional independence assumption, even an **ideally trained model** suffers unavoidable degradation under parallel decoding, and the severity grows with $C(Y|X)$. Thus, while *Sudoku* ($C(Y|X)=0$) is solvable in a single step for an ideal model, *Latin Square* ($C(Y|X)>0$) would still degrade under parallel decoding.
>
> However, current dLLMs such as LLaDA are **not ideal on these tasks**, so *Latin Square* may appear easier than *Sudoku* simply **due to capacity limitations**.
>
> Crucially, **the theoretical trend still holds**. As parallelism increases, tasks with larger $C(Y|X)$ degrade more severely. Although this is less visible in [Figure 3](https://anonymous.4open.science/r/parben/f03.png), [Figure 6](https://anonymous.4open.science/r/parben/f06.png) clearly places *Latin Square* to the right of *Sudoku*, confirming that *Latin Square* is harder to parallelize.
>
> To further address the concern, we isolated model capacity by **fine-tuning LLaDA 1.5 on both tasks** (see [Figure 17](https://anonymous.4open.science/r/parben/f17.png)). After fine-tuning, both achieve near-perfect one-by-one decoding accuracy, but as parallelism increases, *Sudoku* remains stable while *Latin Square* degrades sharply, exactly as predicted by theory.
>
> In summary, although current results on *Sudoku* and *Latin Square* are affected by limited model capacity, we expect future models with sufficient capacity to follow the theoretical trend, and **we include both tasks to support future research**.
>
> ---
>
> > **W2.1: The empirical results for 3f-j show accuracy going down as the # tokens per step increase, which is to be expected that the more naive parallelism (as in the technique does not explicitly target/decide when parallelism is appropriate) employed the worse the accuracy would be.**
>
> Thank you for the thoughtful point. We agree that accuracy naturally decreases as the number of tokens per step increases, and most realistic tasks indeed follow this behavior.
>
> What we emphasize, however, is that the degradation rate differs significantly across tasks. Even in Figures 3 (f-j), all tasks show declining accuracy as parallelism increases, but the steepness of the decline varies, revealing clear differences in task-specific parallelizability. This variation is precisely what [Figure 6](https://anonymous.4open.science/r/parben/f06.png) highlights.
>
> As shown in [Figures 6](https://anonymous.4open.science/r/parben/f06.png) and [33](https://anonymous.4open.science/r/parben/f33.png), existing benchmarks such as GSM8K, IFEval, MATH, and HumanEval are all located in the easy-to-parallelize region and show little separation in difficulty. Developing intelligent parallel decoding methods by evaluating them only on these benchmarks risks overfitting to this narrow range of parallelization difficulty.
>
> In contrast, ParallelBench covers a broad spectrum from very easy-to-parallelize to very hard-to-parallelize. Developing an intelligent parallel decoding method, therefore, requires using high parallelism on easy tasks and reducing parallelism on tasks with strong dependencies. This broad coverage is what allows ParallelBench to test whether a method can genuinely adapt its parallelism to the difficulty of each task.
>
> ---
>
> > **W2.2: Most realistic tasks should fall into these regimes, rather than the extremes of Copy or Replace Index.**
>
> We emphasize that high parallelism is not only useful for synthetic tasks like *Copy* but is also important in many realistic settings. A common example is editing long documents or code, where most of the context is copied unchanged and only a small part is modified. In such cases, using high parallelism for the copying phase provides substantial efficiency gains, whereas relying on one-by-one decoding would give up much of the benefit.
>
> Thus, for easy-to-parallelize parts of real tasks, it is important that an intelligent parallel decoding method can automatically use high parallelism and reduce it only when necessary.

---

> > ### Author Response · Authors · 2025-11-20
> >
> > > **W2.3: It is unlikely to be possible to perform the same information-theoretic analysis on realistic tasks to quantify the theoretic accuracy.**
> >
> > Thank you for this question. As explained in `Major Update 1 (To AC and All Reviewers)`, **our goal is not to model all real-world scenarios**, but to begin with simple, analytically tractable settings where we can measure the inevitable degradation that even an ideally trained model faces under parallel decoding, thereby providing **quantitative insight** into what future models must ultimately overcome.
> >
> > This theoretical understanding was also **essential for constructing ParallelBench** with a broad coverage of parallelization difficulty.
> >
> > Without such grounding, constructing tasks that span different levels of parallelization difficulty would inevitably require ad-hoc, inefficient, and unreliable trial-and-error.
> >
> > Guided by our theoretical analysis, we successfully extended these tasks to realistic settings to span a wide range of parallelization difficulty supported by our experiments.
> >
> > ---
> >
> > > **W3: The difficult task is knowing when parallelism can be enabled, which the work does not provide directions towards any solutions.**
> >
> > As you correctly pointed out, the difficult task is knowing when parallelism can be enabled, and this requires two complementary research directions:
> > 1. Developing intelligent methods that adaptively control parallelism, and
> > 2. Conducting an analysis and building a benchmark to rigorously evaluate whether such methods truly make intelligent parallelism decisions.
> >
> > We view both directions as equally important, and our contribution is (2), providing the evaluation foundation needed for future methods.
> >
> > Therefore, rather than proposing a new method, we evaluated recently proposed methods using ParallelBench to address this concern (see `Major Update 4` in `To AC and All Reviewers`).
> >
> > To accelerate progress toward truly efficient dLLMs, we will continue to integrate newly released parallel decoding methods into ParallelBench and regularly report their speed–quality trade-offs.
> >
> > ---
> >
> > > **Q2: On Figure 7, what is the arrow indicative of?**
> >
> > The arrow in Figure 7 shows that the oracle can greatly increase parallelism while preserving top accuracy. This indicates the need for more intelligent method development, and ParallelBench is designed to drive progress on this open problem.
> >
> > ---
> >
> > > **Q3: Is there a reason why ParallelBench is for evaluating dLLM specifically and not parallel decoding in different LLMS (e.g. parallel decoding in AR LLMs) in general?**
> >
> > Thank you for the insightful question. As you noted, ParallelBench can serve as a general evaluation framework for all parallel decoding models; we focus on dLLMs simply because they are currently the most representative models using parallel decoding.
> >
> > In response, we additionally evaluate parallel decoding in autoregressive LLMs using ParallelBench by testing Medusa [1]. Although Medusa was originally designed for speculative decoding, it is possible to perform parallel decoding by skipping the verification step.
> >
> > As shown in [Figure 32](https://anonymous.4open.science/r/parben/f32.png), parallel decoding leads to quality degradation across all tasks, and Medusa’s limited capacity further contributes to this trend. For example, even the *Copy* task shows substantial accuracy drops as parallelism increases.
> >
> > We believe ParallelBench will likewise provide a solid foundation for advancing parallel decoding in autoregressive LLMs.
> >
> > [1] Medusa: Simple LLM Inference Acceleration Framework with Multiple Decoding Heads
> >
> > ---
> >
> > **Final Note:** We appreciate your feedback and believe we have addressed all concerns by providing additional results and clarification on the *Sudoku* vs. *Latin Square* case, as well as further explanation of our contributions and benchmark scope. If our responses have adequately addressed your concerns, we would greatly appreciate it if you would consider raising your score and supporting our paper’s acceptance. Please feel free to reach out if you have any further questions.
> >
> > We appreciate your time and look forward to your response.

---

> > > ### Comment · Reviewer_vHaA · 2025-11-24
> > >
> > > Thank you for the answers and clarifications.
> > >
> > > W1 Q1: Thank you for the clarification. The additional results from Figure 17 addresses my concern.
> > >
> > > W2.1, W3: Thank you for the clarification. I may not have expressed my concern well in the review. I had meant that the work lacked evaluation on parallel decoding methods, which would be more realistic than naive parallelism and would more strongly show the benefits of the benchmarks on evaluating the capabilities of parallel decoding methods. The additional results in `Appendix L` address this concern. I believe Fig 3 should be updated with the performance of these existing methods, since it demonstrates the value of the benchmarks more than the naive methods.
> > >
> > > W2.2-2.3, Q2: Thank you for the clarification and insights. My concerns are addressed.
> > >
> > > Q3: Could the authors clarify what "most representative models using parallel decoding" means? There has been extensive work on enabling parallel decoding in autoregressive models[1-7]. While it is not expected that this work evaluates the benchmarks on all parallel decoding methods that exist, it is curious that the work is presented as strictly a dLLM benchmark when the insights should apply to all parallel decoding methods. I believe the reason for the narrow scope should be discussed in the conclusion.
> > >
> > > [1] Xuefei Ning, Zinan Lin, Zixuan Zhou, Zifu Wang, Huazhong Yang, and Yu Wang. Skeleton-of-thought: Prompting llms for efficient parallel generation. ICLR, 2025.
> > >
> > > [2] Mingdao Liu, Aohan Zeng, Bowen Wang, Peng Zhang, Jie Tang, and Yuxiao Dong. APAR: Llms can do auto-parallel auto-regressive decoding. arXiv preprint arXiv:2401.06761, 2024.
> > >
> > > [3] Tian Jin, Ellie Y Cheng, Zack Ankner, Nikunj Saunshi, Blake M Elias, Amir Yazdanbakhsh, Jonathan Ragan-Kelley, Suvinay Subramanian, and Michael Carbin. Learning to keep a promise: Scaling language model decoding parallelism with learned asynchronous decoding. ICML, 2025.
> > >
> > > [4] Xinyu Yang, Yuwei An, Hongyi Liu, Tianqi Chen, and Beidi Chen. Multiverse: Your language models secretly decide how to parallelize and merge generation. arXiv preprint arXiv:2506.09991, 2025.
> > >
> > > [5] Gleb Rodionov, Roman Garipov, Alina Shutova, George Yakushev, Erik Schultheis, Vage Egiazarian, Anton Sinitsin, Denis Kuznedelev, and Dan Alistarh. Hogwild! inference: Parallel llm generation via concurrent attention. NeurIPS, 2025.
> > >
> > > [6] Hao Wen, Yifan Su, Feifei Zhang, Yunxin Liu, Yunhao Liu, Ya-Qin Zhang, and Yuanchun Li. Parathinker: Native parallel thinking as a new paradigm to scale llm test-time compute. arXiv preprint arXiv:2509.04475, 2025.
> > >
> > > [7] Jiayi Pan, Xiuyu Li, Long Lian, Charlie Snell, Yifei Zhou, Adam Yala, Trevor Darrell, Kurt Keutzer, and Alane Suhr. Learning adaptive parallel reasoning with language models. COLM, 2025.

---

> > > > ### Author Response · Authors · 2025-11-25
> > > >
> > > > Thank you for your response. We are glad that our rebuttal addresses your concerns. We have **updated Fig. 3** following your suggestion.
> > > >
> > > > > **Q3: Could the authors clarify what "most representative models using parallel decoding" means? There has been extensive work on enabling parallel decoding in autoregressive models [1-7].**
> > > >
> > > > Thank you for raising this important clarification. We appreciate the comprehensive list of related works [1–7].
> > > >
> > > > To clarify our intent: ParallelBench focuses on **token-level parallelism**, where a model attempts to generate multiple tokens within a single sequence in parallel.
> > > >
> > > > In contrast, the parallel decoding methods for AR-LLMs cited in [1–7] primarily implement **task-level parallelism**. These approaches decompose a problem into subtasks and then generate each subtask's content in parallel. For instance, as demonstrated in APAR [2], given a question such as *"What if I can't sleep at night?"*, an LLM may construct subtasks (e.g., *1. Establish a Routine, 2. Create a Restful Environment, 3. Manage Daily Stress*) and complete the content for each subtask in parallel. However, each parallel branch still generates sentences through **one-by-one decoding**.
> > > >
> > > > For instance, in the task of paraphrasing a single sentence:
> > > > * **dLLMs** (token-level parallelism): Can attempt to generate multiple tokens in parallel to produce a paraphrased sentence.
> > > > * **AR-LLMs** with **token-level parallelism** (e.g., Medusa): Can also generate multiple tokens in parallel.
> > > > * **AR-LLMs** with **task-level parallelism** [1–7]: Because the task consists of a single, non-decomposable sequence, these methods cannot parallelize it and inevitably fall back to one-by-one decoding.
> > > >
> > > > In summary, our work focuses on token-level, rather than task-level, parallelism, and we accordingly evaluate dLLMs, which are the primary class of models explicitly designed to support token-level parallel decoding. By contrast, AR-LLMs are not architecturally designed for such parallelism, and recent research [1–7] has naturally focused on parallelism at the sub-task level instead. Nevertheless, we also provide experiments using Medusa, which enables token-level parallel decoding on AR-LLMs.
> > > >
> > > > For completeness, we additionally provide qualitative samples showing that APAR [2], Multiverse [4], and Hogwild [5] do not perform token-level parallel decoding on ParallelBench.
> > > >
> > > > * **APAR** on the *Words-to-Sentence* task (see [Figure](https://anonymous.4open.science/r/parben/task_apar.png)): As the task is non-decomposable, APAR fails to split the task into subtasks and just outputs the final answer without employing parallelism.
> > > > * **Multiverse** on the *Paraphrasing* task (see [Figure](https://anonymous.4open.science/r/parben/task_multiverse.png)): Multiverse uses the `<Parallel>` tag to generate the content of each subtask in parallel. However, in the paraphrasing task, the input consists of a single sentence that cannot be decomposed into independent subtasks. As a result, Multiverse does not activate the `<Parallel>` tag and instead produces the output using standard one-by-one decoding.
> > > > * **Hogwild** on the *Copy* task (see [Figure](https://anonymous.4open.science/r/parben/task_hogwild.png)): Hogwild coordinates multiple LLMs (Alice and Bob) that communicate and divide work across subtasks. In the Copy task, the input is a single, non-decomposable sequence. Bob takes the role of verifying Alice's output suggestions; however, this is inefficient given the simplicity of the task. Consequently, the method offers no advantage from parallel generation and fails to achieve token-level parallelism.
> > > >
> > > > As shown, **task-level parallelism** methods do not perform token-level parallel decoding on ParallelBench because the tasks in ParallelBench are simple and non-decomposable. As a result, these methods revert to **one-by-one decoding** and **cannot achieve parallelism on our benchmark**.
> > > >
> > > > Following your suggestion, we have **updated the Conclusion section** in the updated paper. Details are in `Appendix P`.
> > > >
> > > > ---
> > > >
> > > > Thank you again for your valuable feedback, which has helped us strengthen the paper. Please let us know if there are any remaining questions. We look forward to your response.

---

> > > > > ### Comment · Reviewer_vHaA · 2025-11-25
> > > > >
> > > > > Thank you for the detailed clarification. This has been greatly insightful. My concerns are addressed. I will update my score.

---

> > > > > > ### Author Response · Authors · 2025-11-26
> > > > > >
> > > > > > We sincerely thank the reviewer for their time, thoughtful engagement, and the positive update to the score.

---

### Official Review · Reviewer_o6Fd · 2025-10-31

**Soundness:** 3
**Presentation:** 3
**Contribution:** 3
**Rating:** 6
**Confidence:** 2

**Summary:**

The paper introduces ParallelBench, a benchmark and analysis suite for diffusion LLMs (dLLMs) that isolates the speed--quality trade-offs of parallel decoding. It provides an information-theoretic lower bound (via conditional total correlation) showing why parallel decoding degrades when token dependencies are strong, confirms the theory on analytically tractable list operations, and then demonstrates the same failure modes on realistic tasks across three categories (Waiting Line, Text Writing, Puzzles). The study further contrasts static (Top-k) and adaptive (Threshold) unmasking, semi-AR strategies, and "oracle" per-sample thresholds, revealing substantial headroom for adaptive methods.

**Strengths:**

1. The theory connects nicely to practice: the conditional-correlation argument predicts the “New City” style errors you later see in real tasks. It’s satisfying when the math lines up with the plots.

2. The benchmark fills a gap. We've all seen parallel decoding look great in one demo and fall apart in another; this suite finally gives a way to measure where it cracks and by how much.

3. The oracle threshold curves are especially helpful -- they show there’s real headroom for adaptive, per-sample control rather than one-size-fits-all knobs.

4. The write-up is clear and the figures are readable; I didn’t have to reverse-engineer the setup to follow the argument.

**Weaknesses:**

1. Coverage feels a bit narrow in the main text. A compact table that pulls model/method results out of the appendix would make the big picture easier to scan.

2. The quality metrics are fine, but a couple more "at a glance" indicators (e.g., BERTScore or constraint-violation counts) would help interpret how quality degrades as parallelism goes up.

3. Most examples are short sequences. One longer-form scenario in the main paper would help readers judge whether the conclusions hold in more realistic lengths.

**Questions:**

1. You show big gaps between fixed thresholds and the oracle per-sample threshold. How close can a simple learned heuristic (say, a tiny classifier on token stats) get to the oracle without heavy training?

2. The task taxonomy is useful. Could you add a small table of "meta-features" (rough dependency strength, locality vs. global constraints, etc.) so practitioners can guess which decoding policy to use before running heavy tests?

3. Semi-AR helps sometimes and hurts other times. Do you envision a lightweight runtime policy that flips block sizes per sample using quick signals (entropy, margin, "AR-ness")? Any early results there?

---

> ### Author Response · Authors · 2025-11-20
>
> We appreciate the reviewer’s positive feedback, especially for noting that (i) our theoretical analysis connects nicely to practice, (ii) our proposed benchmark fills a gap, (iii) we show that there is real headroom for more intelligent parallel decoding methods, and (iv) the paper is clearly written.
>
> ***
>
> > **W1: A compact table that pulls model/method results out of the appendix would make the big picture easier to scan.**
>
> Thank you for this valuable suggestion.
>
> We followed your suggestion and added a compact "big-picture" table (see [Table 2](https://anonymous.4open.science/r/parben/t02.png)) to the main body. This table provides an at-a-glance overview of how different decoding methods perform across dLLM models as the number of tokens per step increases.
>
> ---
>
> > **W2: The quality metrics are fine, but a couple more "at a glance" indicators (e.g., BERTScore or constraint-violation counts) would help interpret how quality degrades as parallelism goes up.**
>
> Thank you for the suggestion. As described in [Table 3](https://anonymous.4open.science/r/parben/t03.png), we have included both BERTScore and constraint-violation–style metrics into our evaluation. Specifically, for the *Text Writing* category, we use ROUGE-L for *Summarization*, BERTScore and (1 − BLEU) for *Paraphrasing*, and an inclusion-based accuracy metric (a constraint-violation measure) for *Words-to-Sentence*.
>
> For completeness, we present the experimental results for ROUGE-L, BERTScore, and constraint-violation metrics (See [Figure 29](https://anonymous.4open.science/r/parben/f29.png)).
>
> ---
>
> > **W3: Most examples are short sequences. One longer-form scenario in the main paper would help readers judge whether the conclusions hold in more realistic lengths.**
>
> Thank you for pointing this out.
>
> As stated in Section 6, we intentionally focus on short-output settings to isolate the effects of parallel decoding from model capacity limitations. In practice, even SOTA dLLMs such as LLaDA and Dream achieve low one-by-one decoding accuracy on *Waiting Line* tasks when n increases, despite these tasks being conceptually simple. This indicates that longer outputs quickly introduce capacity bottlenecks.
>
> Importantly, ParallelBench is designed so that **output length can be freely adjusted through hyperparameters**, enabling longer-output evaluations as future, higher-capacity dLLMs become available.
>
> In response, we report results on longer outputs (see [Figure 27](https://anonymous.4open.science/r/parben/f27.png) and [28](https://anonymous.4open.science/r/parben/f28.png)). For *Waiting Line*, *Paraphrasing*, and *W2S*, we increase each task’s core hyperparameters and scale the max generation length by 4×. Across all tasks, accuracy rapidly converges to near-zero as parallelism increases. This suggests that limited model capacity significantly amplifies parallel decoding degradation for longer outputs. Details are in `Appendix J` of the updated paper.

---

> > ### Author Response · Authors · 2025-11-20
> >
> > > **Q1: How close can a simple learned heuristic (say, a tiny classifier on token stats) get to the oracle without heavy training?**
> >
> > Thank you for your valuable suggestion. We found that several recently released parallel decoding methods [1,2,3] share a similar spirit with your idea, in that they improve the parallel decoding strategy using heuristic-based approaches.
> >
> > Therefore, in response, we additionally evaluated methods [1,2,3] on ParallelBench (see [Figure 30 (a)](https://anonymous.4open.science/r/parben/f30.png)).
> >
> > While methods [1,2] show worse trade-off curves than the *Confidence Threshold* method on ParallelBench, method [3] achieves a superior curve, marking a step forward for intelligent parallel decoding. Ultimately, all tested methods remain well below the oracle, suggesting considerable potential for further research.
> >
> >
> > To accelerate progress toward truly efficient dLLMs, we will continue to integrate newly released parallel decoding methods into ParallelBench and regularly report their speed–quality trade-offs.
> >
> > [1] Accelerating Diffusion Large Language Models with SlowFast Sampling: The Three Golden Principles
> > [2] Plan for Speed: Dilated Scheduling for Masked Diffusion Language Models
> > [3] Wide‑In, Narrow‑Out: Revokable Decoding for Efficient and Effective DLLMs
> >
> > ---
> >
> > > **Q3: Do you envision a lightweight runtime policy that flips block sizes per sample using quick signals (entropy, margin, "AR-ness")?**
> >
> > We found that the recently released parallel decoding method, Adaptive Parallel Decoding (APD) [1], shares a similar spirit with your suggestion. APD generates outputs left-to-right while adaptively adjusting the block size at each decoding step. Motivated by this, we further tested APD on ParallelBench.
> >
> > We report results on Dream since APD currently supports only Dream. As shown in [Figure 30 (b)](https://anonymous.4open.science/r/parben/f30.png), APD presents mixed results, outperforming *Confidence Threshold* at high parallelism but falling behind at low parallelism, suggesting considerable room for further improvement.
> >
> > [1] Accelerating Diffusion LLMs via Adaptive Parallel Decoding
> >
> > ---
> >
> > > **Q2: Could you add a small table of "meta-features" (rough dependency strength, locality vs. global constraints, etc.) so practitioners can guess which decoding policy to use before running heavy tests?**
> >
> > Thank you for the valuable suggestion. Following your recommendation, we have added a small table of meta-features in [Tables 6](https://anonymous.4open.science/r/parben/t06.png) and [7](https://anonymous.4open.science/r/parben/t07.png) of `Appendix G.1`.
> >
> > ---
> >
> > **Final Note:** Thank you for your valuable comments. We are grateful that you found our theory, benchmark, and evaluations valuable. If there are any remaining questions, please do not hesitate to let us know. Assuming our responses have satisfactorily addressed your concerns, we kindly ask you to consider enhancing your score and supporting our paper.
> >
> > We look forward to your response.

---

### Official Review · Reviewer_uEfF · 2025-10-31

**Soundness:** 3
**Presentation:** 3
**Contribution:** 3
**Rating:** 4
**Confidence:** 2

**Summary:**

his paper investigates the speed–quality trade-off in parallel decoding for Diffusion Language Models (dLLMs). Although dLLMs promise faster inference through parallel decoding, the underlying conditional independence assumption often leads to severe quality degradation in tasks with strong token dependencies. To expose this issue, the authors provide an information-theoretic analysis and introduce ParallelBench, specifically designed for evaluating dLLMs under parallel decoding. It comprises 17 tasks across three categories (Waiting Line, Text Writing, Puzzles) that are easy for humans and autoregressive (AR) models but challenging for dLLMs. Experiments reveal that (1) dLLMs experience substantial quality loss during parallel decoding, and (2) existing decoding strategies (both static and adaptive) fail to balance quality and speed effectively.

**Strengths:**

1. This paper tackles a core dLLM challenge: the quality impact of parallel decoding, a key advantage over AR models, and highlights the limitations of existing benchmarks in assessing it.
2. Provides theoretical insights.
3. Clearly illustrates token dependency variations across tasks through analysis of synthetic “list operations” such as Copy, Replace Index, Replace Random, and Shuffle.
4. Systematically evaluates multiple dLLMs—including LLaDA, Dream, and the closed-source Mercury—across various decoding strategies such as Top-k, Threshold, and Semi-AR on PARALLELBENCH.

**Weaknesses:**

1. Need for benchmark comparison: More experiments are needed to show PARALLELBENCH’s added value over existing benchmarks.
2. Missing actual speed measurements: Experiments focus on parallelism vs. quality, but no wall-clock latency or time–quality curves are provided.
3. Limited real-world coverage: The benchmark may not capture all challenges dLLMs face in complex, open-ended tasks.

**Questions:**

See Weaknesses

---

> ### Author Response · Authors · 2025-11-20
>
> We thank the reviewer for the thoughtful comments, in particular for noting that (i) our paper tackles a core dLLM challenge, (ii) it provides theoretical insights, (iii) our synthetic studies clearly illustrate task-dependent token dependencies, and (iv) it systematically evaluates multiple dLLMs across various decoding strategies.
>
> ---
>
> > **W1: Need for benchmark comparison: More experiments are needed to show PARALLELBENCH’s added value over existing benchmarks.**
>
> Thank you for this important question. ParallelBench offers two major components that make it distinct from existing benchmarks, further supported by additional experiments on MATH [1] and IFEval [2] shown below.
>
> **1. Simple but Hard-to-Parallelize Tasks**
>
> Unlike existing benchmarks such as GSM8K, we design simple tasks that achieve near-perfect one-by-one decoding accuracy but suffer severe degradation under parallel decoding due to the conditional independence assumption, even for ideally trained models. ParallelBench captures exactly these stress-test scenarios.
>
> **2. Broad Coverage of Parallelization Difficulty**
>
> We showed in [Figure 6](https://anonymous.4open.science/r/parben/f06.png) that existing benchmarks such as GSM8K and HumanEval cover only a narrow region of the easy-to-parallelize spectrum. To further validate this observation, **we additionally analyzed MATH and IFEval**, which likewise span only a narrow portion of the difficulty spectrum. In contrast, ParallelBench spans a much wider range, from very easy-to-parallelize tasks, such as Copy, to very hard-to-parallelize tasks, such as Shuffle.
>
> Developing new parallel decoding methods only on a narrow range of benchmarks risks overfitting to a small range of parallelization difficulty. In contrast, ParallelBench enables rigorous evaluation of whether a method can intelligently adjust its degree of parallelism from very low to very high depending on task difficulty, thereby supporting the development of truly efficient dLLMs.
>
> [1] Measuring Mathematical Problem Solving With the MATH Dataset
> [2] Instruction-Following Evaluation for Large Language Models
>
> ---
>
> > **W2: Missing actual speed measurements: Experiments focus on parallelism vs. quality, but no wall-clock latency or time–quality curves are provided.**
>
> Thank you for this question. Since our goal is to study the speed–quality trade-off of parallel decoding methods, we compare different decoding strategies while keeping the underlying dLLM model fixed, ensuring fair and controlled evaluation. For a fixed model, per-step inference time is constant, so wall-clock latency is directly determined by the number of decoding steps, and effective speed scales with the number of tokens unmasked per step. We therefore use tokens per timestep as a hardware-agnostic speed metric, avoiding wall-clock measurements that depend heavily on hardware and runtime environments.
>
> For completeness, we also provide actual wall-clock latency measurements and their corresponding time–quality curves below. We obtained these results using a single A100 80GB GPU with a batch size of 1. As shown in [Figure 31](https://anonymous.4open.science/r/parben/f31.png), the time-quality curve resembles a horizontally mirrored version of our speed-quality curve, which uses the number of tokens per step on the x-axis.

---

> > ### Author Response · Authors · 2025-11-20
> >
> > > **W3: Limited real-world coverage: The benchmark may not capture all challenges dLLMs face in complex, open-ended tasks.**
> >
> > While we agree that ParallelBench may not capture all aspects of real-world tasks, this design choice follows the same philosophy used by existing benchmarks like RULER [1] and IFEval [2]. Although these benchmarks are far from realistic scenarios, they deliberately adopt simplified environments to isolate particular capabilities, and they are widely recognized as valuable contributions.
> >
> > |Benchmark|Target Capability|Tasks|
> > |-|-|-|
> > |RULER|Long-context modeling|Synthetic long sequences; needle-in-a-haystack retrieval|
> > |IFEval| Instruction-following|Unnatural, rule-heavy prompts (e.g., include specific letters)|
> > |**ParallelBench** (ours)|Parallel decoding| *Waiting Line, Text Writing, Puzzles*|
> >
> > Similarly, while we design our tasks to be as realistic as possible, our primary goal is to stress-test parallel decoding by exposing when the conditional independence assumption breaks down. Just as RULER and IFEval have advanced our understanding of long-context modeling and instruction-following, ParallelBench provides a similarly valuable foundation for developing intelligent parallel decoding methods in dLLMs.
> >
> > [1] RULER: What's the Real Context Size of Your Long-Context Language Models?
> > [2] Instruction-Following Evaluation for Large Language Models
> >
> > ***
> >
> > **Final Note:** We appreciate your thoughtful feedback and are glad that you recognized the strengths of our work. We have added comparisons with existing benchmarks, provided speed measurements, and clarified the scope of real-world coverage in response to your comments. If you find that we have fully addressed your concerns, we would be grateful if you would consider adjusting your score accordingly.
> >
> > We appreciate your time and look forward to your response.

---

### Official Review · Reviewer_WAQL · 2025-11-01

**Soundness:** 3
**Presentation:** 3
**Contribution:** 3
**Rating:** 6
**Confidence:** 4

**Summary:**

The paper studies parallel decoding in diffusion-based large language models (dLLMs) in order to address the challenge that parallel decoding often fails to capture inter-token dependencies leading to quality degradation, due to the conditional independence assumption among tokens.

The authors make three main contributions:

1. Information-theoretic analysis — They formalize the lower bound of parallel decoding quality loss using conditional total correlation C(Y|X), proving that even ideal models face an inherent speed–quality trade-off.

2.	Synthetic case studies — They analyze list operations like Copy, Replace Random, and Shuffle to quantify how dependency strength affects parallel decoding accuracy.

3.	PARALLELBENCH — A new benchmark with 17 tasks across 3 categories (Waiting Line, Text Writing, Puzzles) to empirically evaluate dLLMs and AR LLMs. It exposes how parallel decoding degrades quality in realistic settings and shows that current adaptive unmasking methods (e.g., Top-k, Threshold) cannot fully balance speed and accuracy.

The paper concludes that dLLMs suffer from severe quality degradation under parallel decoding, especially in dependency-heavy tasks, and that current decoding strategies fail to adapt parallelism dynamically to task difficulty.

**Strengths:**

1. The use of conditional total correlation to quantify unavoidable quality loss provides a solid mathematical basis for analyzing the parallel decoding trade-off, making the paper has clear theoretical grounding.

2. The combination of theoretical proofs, synthetic tasks, and realistic benchmarks (including grammar-sensitive and reasoning tasks) provides a well-rounded evaluation.

3. The combination of theoretical proofs, synthetic tasks, and realistic benchmarks (including grammar-sensitive and reasoning tasks) provides a well-rounded evaluation.

4. The writing, presentations accompanied with interpretations are overall clear. Code is publicly released, and the benchmark tasks are well-documented, facilitating future research.

**Weaknesses:**

1. Although 17 tasks span diverse categories, most involve short outputs or synthetic patterns; results may not generalize to long-context reasoning or dialogue generation.

2. Comparisons largely focus on fixed unmasking or basic threshold schemes, missing newer adaptive scheduling approaches (e.g., dilated scheduling, SlowFast decoding, or hybrid AR-diffusion).

3. The paper focuses on decoding strategies, not how model pre-training or architecture might affect the issue.

4. Other decoding/scheduling work that should be covered (and possibly compared): the paper could be strenghthen to include decoding strategies that cover dynamic stage-based scheduling [1], dilated unmasking[2], block decoding [3,5] and revocation/remasking [4].



-----
References
1. Accelerating Diffusion Large Language Models with SlowFast Sampling: The Three Golden Principles (Wei et al., 2025)
2. Plan for Speed: Dilated Scheduling for Masked Diffusion Language Models (Luxembourg, Permuter, Nachmani, 2025)
3. Fast‑dLLM: Training‑free Acceleration of Diffusion LLM by Enabling KV Cache and Parallel Decoding
4. Wide‑In, Narrow‑Out: Revokable Decoding for Efficient and Effective DLLMs (Hong et al., 2025)
5., Block Diffusion: Interpolating Between Autoregressive and Diffusion Language Models (Arriola et al., 2025)

**Questions:**

1. The oracle analysis suggests per-sample thresholds could yield large gains — how feasible is this in practical inference (e.g., latency, calibration)?

2. How do results change with larger blocks or variable block scheduling strategies beyond fixed lengths (with semi-AR)?

---

> ### Author Response · Authors · 2025-11-20
>
> We thank the reviewer for the encouraging feedback, especially for recognizing that (i) our paper provides a solid mathematical basis with clear theoretical grounding, (ii) our evaluation is well-rounded, combining theory, synthetic tasks, and realistic benchmarks, (iii) the writing and presentation are clear, and (iv) our public code and well-documented benchmark will facilitate future research.
>
> ***
>
> > **W1.1: Although 17 tasks span diverse categories, most involve short outputs.**
>
> We thank the reviewer for pointing this out.
>
> As stated in Section 6, we intentionally focus on short-output settings to isolate the effects of parallel decoding from model capacity limitations. In practice, even SOTA dLLMs such as LLaDA and Dream achieve low one-by-one decoding accuracy on *Waiting Line* tasks when n increases, despite these tasks being conceptually simple. This indicates that longer outputs quickly introduce capacity bottlenecks.
>
> Importantly, ParallelBench is designed so that **output length can be freely adjusted through hyperparameters**, enabling longer-output evaluations as future, higher-capacity dLLMs become available.
>
> In response, we report results on longer outputs (see [Figure 27](https://anonymous.4open.science/r/parben/f27.png) and [28](https://anonymous.4open.science/r/parben/f28.png)). For *Waiting Line*, *Paraphrasing*, and *W2S*, we increase each task’s core hyperparameters and scale the max generation length by 4×. Across all tasks, accuracy rapidly converges to near-zero as parallelism increases. This suggests that limited model capacity significantly amplifies parallel decoding degradation for longer outputs. Details are in `Appendix J` of the updated paper.
>
> ***
>
> > **W1.2: Synthetic patterns; Results may not generalize to long-context reasoning or dialogue generation.**
>
> While we agree that ParallelBench may not generalize to long-context reasoning or dialogue generation, this design choice follows the same philosophy used by existing benchmarks like RULER [1] and IFEval [2]. Although these benchmarks are far from realistic scenarios, they deliberately adopt simplified environments to isolate particular capabilities, and they are widely recognized as valuable contributions.
>
> |Benchmark|Target Capability|Tasks|
> |-|-|-|
> |RULER |Long-context modeling|Synthetic long sequences; needle-in-a-haystack retrieval|
> |IFEval | Instruction-following|Unnatural, rule-heavy prompts (e.g., include specific letters)|
> |**ParallelBench** (ours)|Parallel decoding| *Waiting Line, Text Writing, Puzzles*|
>
> Similarly, while we design our tasks to be as realistic as possible, our primary goal is to stress-test parallel decoding by exposing when the conditional independence assumption breaks down. Just as RULER and IFEval have advanced our understanding of long-context modeling and instruction-following, ParallelBench provides a similarly valuable foundation for developing intelligent parallel decoding methods in dLLMs.
>
> [1] RULER: What's the Real Context Size of Your Long-Context Language Models?
> [2] Instruction-Following Evaluation for Large Language Models
>
> ***
>
> > **W3: The paper focuses on decoding strategies, not how model pre-training or architecture might affect the issue.**
>
> As we theoretically show in Section 4, the conditional independence assumption in dLLM parallel decoding causes inevitable degradation as parallelism increases, even for an ideally trained model. Thus, architectural or pre-training changes do not resolve the fundamental issue we study, so we construct ParallelBench using tasks that are simple enough that model capacity is not the main bottleneck, allowing us to focus on the decoding strategies themselves.
>
> We nevertheless evaluated additional variants, including models with different pre-training (e.g., LLaDA 1.0) and architectures (e.g., Dream and DiffuCoder). As shown in `Appendix I.2`, all exhibit similar qualitative behaviors, confirming that these effects persist across model designs and training recipes.

---

> > ### Author Response · Authors · 2025-11-20
> >
> > > **W2, W4: Missing newer adaptive scheduling approaches (e.g., dilated scheduling, SlowFast decoding, or hybrid AR-diffusion). The paper could be strenghthen to include decoding strategies that cover dynamic stage-based scheduling [1], dilated unmasking[2], block decoding [3,5] and revocation/remasking [4].**
> >
> > Thank you for the valuable suggestion. Following your feedback, we additionally evaluated the methods [1,2,4] using our ParallelBench (see [Figure 30 (a)](https://anonymous.4open.science/r/parben/f30.png)).
> >
> > While methods [1,2] show worse trade-off curves than the *Confidence Threshold* method on ParallelBench, method [4] achieves a superior curve, marking a step forward for intelligent parallel decoding. Ultimately, all tested methods remain well below the oracle, suggesting considerable potential for further research.
> >
> > We remark that we already provide evaluation and analysis of block decoding methods [3,5] in our paper under the name semi-autoregressive decoding. Furthermore, for Fast-dLLM [3], we provide evaluation and analysis of its performance with KV cache in `Appendix E.3`.
> >
> > [1] Accelerating Diffusion Large Language Models with SlowFast Sampling: The Three Golden Principles
> > [2] Plan for Speed: Dilated Scheduling for Masked Diffusion Language Models
> > [3] Fast‑dLLM: Training‑free Acceleration of Diffusion LLM by Enabling KV Cache and Parallel Decoding
> > [4] Wide‑In, Narrow‑Out: Revokable Decoding for Efficient and Effective DLLMs
> > [5] Block Diffusion: Interpolating Between Autoregressive and Diffusion Language Models
> >
> > ***
> >
> > > **Q2: How do results change with larger blocks or variable block scheduling strategies beyond fixed lengths (with semi-AR)?**
> >
> > Thank you for this insightful question. For results regarding larger block sizes, we refer the reviewer to `Appendix E.4`.
> >
> > For variable block scheduling, we also evaluate Adaptive Parallel Decoding (APD) [1]. APD generates outputs left-to-right and adaptively adjusts the block size at each decoding step, sharing a similar spirit to the variable-block scheduling strategy you suggested.
> >
> > We report results on Dream since APD currently supports only Dream. As shown in [Figure 30 (b)](https://anonymous.4open.science/r/parben/f30.png), APD presents mixed results, outperforming *Confidence Threshold* at high parallelism but falling behind at low parallelism, suggesting considerable room for further improvement.
> >
> > [1] Accelerating Diffusion LLMs via Adaptive Parallel Decoding
> >
> > ***
> >
> > > **Q1: The oracle analysis suggests per-sample thresholds could yield large gains — how feasible is this in practical inference (e.g., latency, calibration)?**
> >
> > Oracle performance illustrates the level that an intelligent method could achieve. Reaching this level in practice is difficult, since estimating per-sample difficulty using calibration or early-step confidence patterns must balance latency and stability, making it a nontrivial design problem. This highlights the need for novel decoding methods, and ParallelBench provides a foundation for developing them.
> >
> > While we further tested various recent methods, as shown above, we found that they still lag behind oracle performance. To accelerate progress toward truly efficient dLLMs, we will continue to integrate newly released parallel decoding methods into ParallelBench and regularly report their speed–quality trade-offs.
> >
> > ***
> >
> > **Final Note:** Thank you for your detailed comments. We are glad that you found our theory, benchmark, code, and evaluation helpful for facilitating future research. If there are any remaining questions, please do not hesitate to let us know. If our responses have resolved your concerns, we kindly request that you consider increasing your score and support the acceptance of our paper.
> >
> > We look forward to your response.

---

> > > ### Comment · Reviewer_WAQL · 2025-11-27
> > >
> > > Thank you for providing the responses in addressing my questions. I have updated my score.

---

> > > > ### Author Response · Authors · 2025-11-27
> > > >
> > > > We sincerely appreciate the reviewer's time, valuable feedback, and the positive score update.

---

### Author Response · Authors · 2025-11-20
**To AC and All Reviewers (1/2)**

We want to thank all reviewers for their thoughtful comments and helpful feedback. We are particularly encouraged that the reviewers found that (i) our paper tackles a core challenge (`R-uEfF`), (ii) our paper is clearly written (`R-WAQL`, `R-o6Fd`), (iii) our paper provides solid theoretical analysis (`R-WAQL`, `R-vHaA`) with empirical validation (`R-vHaA`) and meaningful insights (`R-uEfF`, `R-vHaA`), (iv) our theory connects well to practice (`R-o6Fd`) with a well-rounded evaluation (`R-WAQL`), (v) our benchmark is comprehensive (`R-uEfF`, `R-vHaA`) and fills a gap (`R-o6Fd`), and (vi) our publicly released code (`R-WAQL`) and benchmark (`R-WAQL`) facilitate future research (`R-WAQL`, `R-vHaA`) by addressing limitations of existing benchmarks (`R-uEfF`).

In response to the feedback, we addressed each concern, added new results, and updated the paper accordingly, including newly added `Appendices J, K, L, M, N, and O`.

---

## **Major Update 1: Clarification on Real-World Coverage**
While we agree that ParallelBench may not capture all aspects of real-world tasks, this design choice follows the same philosophy used by existing benchmarks like RULER [1] and IFEval [2]. Although these benchmarks are far from realistic scenarios, they deliberately adopt simplified environments to isolate particular capabilities, and they are widely recognized as valuable contributions.

|Benchmark|Target Capability|Tasks|
|-|-|-|
|RULER |Long-context modeling|Synthetic long sequences; needle-in-a-haystack retrieval|
|IFEval | Instruction-following|Unnatural, rule-heavy prompts (e.g., include specific letters)|
|**ParallelBench** (ours)|Parallel decoding| *Waiting Line, Text Writing, Puzzles*|

Similarly, while we design our tasks to be as realistic as possible, our primary goal is to stress-test parallel decoding by exposing when the conditional independence assumption breaks down. Just as RULER and IFEval have advanced our understanding of long-context modeling and instruction-following, we believe that ParallelBench provides a similarly valuable foundation for developing intelligent parallel decoding methods in dLLMs.

[1] RULER: What's the Real Context Size of Your Long-Context Language Models?
[2] Instruction-Following Evaluation for Large Language Models

---

## **Major Update 2: Comparison with Existing Benchmarks**
ParallelBench offers two major components that make it distinct from existing benchmarks.

### **Simple but Hard-to-Parallelize Tasks**
Unlike existing benchmarks such as GSM8K, we design simple tasks that achieve near-perfect one-by-one decoding accuracy but suffer severe degradation under parallel decoding due to the conditional independence assumption, even for ideally trained models. ParallelBench captures exactly these stress-test scenarios.

### **Broad Coverage of Parallelization Difficulty**
We showed in [Figure 6](https://anonymous.4open.science/r/parben/f06.png) that existing benchmarks such as GSM8K and HumanEval cover only a narrow region of the easy-to-parallelize spectrum. To further validate this observation, **we additionally analyzed MATH [1] and IFEval [2]**, which likewise span only a narrow portion of the difficulty spectrum. In contrast, ParallelBench spans a much wider range, from very easy-to-parallelize tasks, such as *Copy*, to very hard-to-parallelize tasks, such as *Shuffle*.

Developing new parallel decoding methods only on a narrow range of benchmarks risks overfitting to a small range of parallelization difficulty. In contrast, ParallelBench enables rigorous evaluation of whether a method can intelligently adjust its degree of parallelism from very low to very high depending on task difficulty, thereby supporting the development of truly efficient dLLMs.

[1] Measuring Mathematical Problem Solving With the MATH Dataset
[2] Instruction-Following Evaluation for Large Language Models

---

> ### Author Response · Authors · 2025-11-20
> **To AC and All Reviewers (2/2)**
>
> ## **Major Update 3: Clarification on the Scope of Our Contributions**
> The fundamental challenge for dLLMs is determining the right degree of parallelism, and this requires two complementary research directions:
> 1. Developing intelligent methods that adaptively control parallelism, and
> 2. Conducting an analysis and building a benchmark to rigorously evaluate whether such methods truly make intelligent parallelism decisions.
>
> We view both directions as equally important, and our contribution is (2), providing the evaluation foundation needed for future methods.
>
> Therefore, rather than proposing a new method, we evaluated recently proposed methods using ParallelBench to address this concern (see `Major Update 4`).
>
> Furthermore, to accelerate progress toward truly efficient dLLMs, we will continue to integrate newly released parallel decoding methods into ParallelBench and regularly report their speed–quality trade-offs.
>
> ---
>
> ## **Major Update 4: Benchmarking New Methods**
> We benchmark the speed–quality trade-offs of the newly released methods [1,2,3,4] (see [Figure 30](https://anonymous.4open.science/r/parben/f30.png)). Since [4] currently supports only Dream, we report results for methods [1,2,3] on LLaDA and for method [4] on Dream.
>
> While methods [1,2] show worse trade-off curves than the *Confidence Threshold* method on ParallelBench, method [3] achieves a superior curve, marking a step forward for intelligent parallel decoding. Method [4] presents mixed results, outperforming *Confidence Threshold* at high parallelism but falling behind at low parallelism. Ultimately, all tested methods remain well below the oracle, suggesting considerable potential for further research.
>
> [1] Accelerating Diffusion Large Language Models with SlowFast Sampling: The Three Golden Principles
> [2] Plan for Speed: Dilated Scheduling for Masked Diffusion Language Models
> [3] Wide‑In, Narrow‑Out: Revokable Decoding for Efficient and Effective DLLMs
> [4] Accelerating Diffusion LLMs via Adaptive Parallel Decoding
>
> ---
>
> ## **Major Update 5: Additional Results on Longer Outputs**
> As stated in `Section 6`, we intentionally focus on short-output settings to isolate the effects of parallel decoding from model capacity limitations. In practice, even SOTA dLLMs such as LLaDA and Dream achieve low one-by-one decoding accuracy on *Waiting Line* tasks when *n* increases, despite these tasks being conceptually simple. This indicates that longer outputs quickly introduce capacity bottlenecks.
>
> Importantly, ParallelBench is designed so that **output length can be freely adjusted through hyperparameters**, enabling longer-output evaluations as future, higher-capacity dLLMs become available.
>
> In response, we report additional results on longer outputs (see [Figure 27](https://anonymous.4open.science/r/parben/f27.png) and [28](https://anonymous.4open.science/r/parben/f28.png)). For *Waiting Line*, *Paraphrasing*, and *W2S*, we increase each task’s core hyperparameters and scale the max generation length by 4×. Across all tasks, accuracy rapidly converges to near-zero as parallelism increases. This suggests that limited model capacity significantly amplifies parallel decoding degradation for longer outputs. Details are in `Appendix J` of the updated paper.

---

### Author Response · Authors · 2025-12-01
**Summary of the Discussion Period**

Dear Area Chair,

Thank you very much for taking the time to handle our submission, especially under the difficult circumstances caused by the recent OpenReview incident.

As you have newly been assigned to our paper, we would like to provide a summary of our discussion period before the reviews were reverted.

---

## **Summary of the Discussion Period**
We provided additional clarifications and experimental results to thoroughly address every concern raised by the reviewers, and we updated the paper accordingly, including the newly added `Appendices J, K, L, M, N, O, and P`.

---

### **`R-vHaA`: Rating 4 → 8, Confidence 3**

We would like to highlight **the constructive and detailed discussion** with `R-vHaA`. Their initial concerns regarding our theoretical analysis were fully resolved through our clarifications and additional experiments. We continued the dialogue as `R-vHaA` asked further follow-up questions, and we offered additional explanations and results. Ultimately, **as documented in the discussion thread,** `R-vHaA` stated that our responses were **"greatly insightful"** and **updated the rating from 4 to 8.**

---

### **`R-WAQL`: Rating 6 → 8, Confidence 4**

The reviewer provided **the highest-confidence (4)** review among all reviewers. They initially rated the paper a 6 and left several questions, to which we responded with clarifications and additional experiments. **As documented in the discussion thread,** `R-WAQL` explicitly stated that our response addressed their questions and subsequently **updated the rating from 6 to 8.**

---

### **`R-o6Fd`: Rating 6, Confidence 2 (No Response)**

The reviewer provided a **positive initial review** along with several suggestions and follow-up questions for strengthening the paper. We addressed these questions through clarifications and additional experiments, and we incorporated all of their suggestions, including:

- Add a compact table into the main text for visibility
- Add an auxiliary metric figure to improve interpretability
- Extend experiments to include long-context settings
- Provide a meta-features table to summarize task characteristics

Unfortunately, the discussion period concluded before `R-o6Fd` had a chance to reply.

---

### **`R-uEfF`: Rating 4, Confidence 2 (No Response)**

We note that this review was self-rated with **low confidence (2)** and provided limited specific critique. We believe all three of `R-uEfF`’s concerns were resolved through our rebuttal with additional clarification and experiments. Unfortunately, the discussion period ended before `R-uEfF` responded.

Here we summarize the concerns raised by `R-uEfF` and how we addressed them:

* Need for benchmark comparison
    * In addition to `Section 6` and `Figure 4`, we added further clarification and additional experiments in response to the reviewer’s request (see `Major Update 2`).
* Missing actual speed measurements
    * We clarified why tokens-per-step is an appropriate hardware-agnostic speed metric under a fixed model and provided additional wall-clock latency and time–quality curves for completeness.
* Limited real-world coverage
    * We provided clarification in `Major Update 1`. This concern was also raised by `R-WAQL`, who acknowledged that our response sufficiently addressed the issue.

---

## **Conclusion**

Before the rollback, the paper held an **average rating of 6.5,** supported by **two ratings of 8** from the **highest-confidence** (4 and 3) reviewers. The only negative score (4) came from the **lowest-confidence** (2) reviewer `R-uEfF`.

Although we provided detailed clarifications and additional experiments that substantially strengthened the paper, the discussion period ended before `R-uEfF` had an opportunity to respond. We hope that our active engagement throughout the discussion will be taken into consideration in your assessment.

Thank you again for your careful evaluation and service to the community.

Sincerely,
The Authors

---

### Meta-Review · Area_Chair_np9c · 2025-12-22

**Summary:**

The paper proposes a metric, the total correlation, as a kind of "difficulty measure" for parallel generation tasks. It constructs several semi-synthetic tasks involving natural tasks like copying, replacing items with chosen values, shuffling and compiles them into a benchmark. They also evaluate several models and decoding strategies on these tasks.

The reviews are guardedly positive. The strengths of the papers are that it fills a gap (o6Fd) in parallel diffusion model evaluation space, nicely abstracts mathematically an intuitive notion of difficulty, and is relatively systematic (uEfF) across several models and decoding strategies. The concerns include how much the results will generalize to natural tasks (WAQL), how much coverage there is in the "failure modes" identified (uEfF). Some of the discussion has resulted in two reviewers raising their score (vHaA and WAQL).

My own opinion is guarded as well. I think having this synthetic benchmark as a sanity check would be good for the community-at-large. On the other hand, I share some of the concerns mentioned: it isn't clear to me how predictive of model performance "in the wild" these synthetic tasks are, and how much coverage they have. I also am somewhat concerned about the metric that's defined in the paper: while they do state the lower bounds on T-step decoding coming from "T-wise" mutual information --- these are never used for the tasks they define. And in fact --- one could easily design tasks that have very small k-wise mutual information, but very large (k-1)-wise mutual information. (In fact, since the focus is on a benchmark of synthetic tasks, one would think the authors would have designed something of this flavor.) Finally, it isn't clear the insights of the synthetic tasks suggest anything about better decoding algorithms or architectures or training procedures.

**Reviewer Concerns:**

Generalization of synthetic tasks to natural tasks, (lack of) coverage of difficulties in natural tasks, somewhat simplistic way of capturing difficulties in parallel decoding.

**Reviewer Scores:**

Two reviewers explicitly stated they would raise their scores (from a 6). Authors in their summary state they raised them to an 8.

---

### Decision · Program_Chairs · 2026-01-26

Accept (Poster)